# Establishment of a morphological atlas of the *Caenorhabditis elegans* embryo using deep-learning-based 4D segmentation

Jianfeng Cao [1,8], Guoye Guan [2,8], Vincy Wing Sze Ho [3,4,8], Ming-Kin Wong [3], Lu-Yan Chan [3], Chao Tang [2,5,6✉], Zhongying Zhao [3,7✉] & Hong Yan [1✉]

The invariant development and transparent body of the nematode *Caenorhabditis elegans* enables complete delineation of cell lineages throughout development. Despite extensive studies of cell division, cell migration and cell fate differentiation, cell morphology during development has not yet been systematically characterized in any metazoan, including *C. elegans*. This knowledge gap substantially hampers many studies in both developmental and cell biology. Here we report an automatic pipeline, CShaper, which combines automated segmentation of fluorescently labeled membranes with automated cell lineage tracing. We apply this pipeline to quantify morphological parameters of densely packed cells in 17 developing *C. elegans* embryos. Consequently, we generate a time-lapse 3D atlas of cell morphology for the *C. elegans* embryo from the 4- to 350-cell stages, including cell shape, volume, surface area, migration, nucleus position and cell-cell contact with resolved cell identities. We anticipate that CShaper and the morphological atlas will stimulate and enhance further studies in the fields of developmental biology, cell biology and biomechanics.

[1] Department of Electrical Engineering, City University of Hong Kong, Hong Kong 999077, China. [2] Center for Quantitative Biology, Peking University, 100871 Beijing, China. [3] Department of Biology, Hong Kong Baptist University, Hong Kong 999077, China. [4] Center for Epigenomics Research, Division of Life Science, Hong Kong University of Science and Technology, Hong Kong 999077, China. [5] Peking-Tsinghua Center for Life Sciences, Peking University, 100871 Beijing, China. [6] School of Physics, Peking University, 100871 Beijing, China. [7] State Key Laboratory of Environmental and Biological Analysis, Hong Kong Baptist University, Hong Kong 999077, China. [8] These authors contributed equally: Jianfeng Cao, Guoye Guan, Vincy Wing Sze Ho. ✉email: tangc@pku.edu.cn; zyzhao@hkbu.edu.hk; h.yan@cityu.edu.hk

Embryogenesis in metazoans involves multidimensional, spatiotemporal cellular changes, including cell proliferation, differentiation, and morphogenesis. The eutelic organism *Caenorhabditis elegans* adopts an invariant developmental trajectory, with reproducible cell lineages and consistent cell division timings, division orientations, cell migration trajectories, and fate differentiations[1]. Therefore, it has been widely used as a model organism for developmental biology research at the cellular level, affording exceptional temporal resolution to such research[2–5]. Previous studies have constructed quantitative *C. elegans* developmental atlases, including atlases of cell division timing[6], gene expression and cell position[7–9], and cell–cell contact mapping and signaling[2,10]. However, due to the lack of an effective cell-membrane marker for the later stages of embryogenesis and a reliable algorithm for the segmentation of time-lapse three-dimensional (3D; hereafter referred to as 4D) images, most of the existing studies have been based on theoretical prediction or modeling, which commonly use nucleus position as a proxy of cell location for cell segmentation. During metazoan embryogenesis, cell morphology is tightly associated with several biological processes, including cell-cycle control[11], spindle formation[12], cell-fate asymmetry and differentiation[13], intercellular signaling[2,14,15], cytomechanics, morphogenesis, and organogenesis[16–18]. However, a precise knowledge of changes in cell morphology during development (e.g., cell shape, cell size, and cell neighborhood) is lacking.

Although recent advances in confocal microscopy have promoted in vivo 4D imaging of the *C. elegans* embryo throughout embryogenesis, the large quantity of volumetric imaging data makes the visual identification of meaningful morphological changes tedious, and the resulting output is not usually quantitative, consequently hampering further functional characterization. To facilitate morphological and functional studies at a cellular resolution, recent studies have highlighted the need for 3D segmentation of cellular surfaces in addition to nuclei[19,20], which considerably reduce the difficulty in analyzing large-scale 4D images. Compared with manual annotation, automatic segmentation can provide objective quantification and improve consistency, reproducibility, and efficiency in defining cell morphology. However, crowded cells and long imaging durations combined with modest image quality due to constraints such as embryo viability, phototoxicity, and photobleaching present a significant challenge for cell segmentation. Unlike nuclei, which are localized and well-separated ellipsoid components, cell membranes are thin planar structures, forming complicated networks. This partially explains why cell-membrane-based segmentation methods are rare, whereas nucleus segmentation and tracing tools, such as StarryNite and AceTree[21,22], are well developed. Additionally, as shown in Supplementary Fig. 1, laser attenuation makes segmentation more challenging for deeper slices. Such problems are aggravated when the membrane is parallel to the focal plane. In theory, a longer exposure duration or a higher laser power may improve the image quality in these cases. However, a careful tradeoff between image quality and phototoxicity has to be made during 4D imaging.

In the past decade, several attempts have been made to boost the performance of membrane surface segmentation. Classical techniques are based on predefined models and image intensity features. Among these, active contour and level set are the two most convincing methods for segmenting images. Active contour methods treat segmentation as an energy minimization process whereby the external image forces push the contours toward object boundaries, whereas internal forces resist the deformation. To mediate the internal and external forces, different evolution equations are utilized to control the deformation process

precisely[23–26]. The level set is designed to diminish the difficulty in finding a desirable representation force by embedding the boundary curve as a real-valued solution to an equation describing the topological features, such as splits and holes. Using coupling constraints in level set evolution, Nath et al.[27] proposed a computationally efficient method to segment hundreds of cells simultaneously. Kiss et al.[28] used level sets to segment plant tissues at multiple scales, effectively reducing the error margin at blurry regions. In practice, however, the implementation of level sets requires considerable computational resources and may fail at incomplete cell boundaries. Xing et al.[29] provided a comprehensive review of classical cell segmentation techniques. For such classical methods, a data-dependent, parameterized preprocessing stage is always required, otherwise the system would be prone to under- or over-segmentation errors.

Recently, deep-learning-based methods have been identified as promising tools for recognition tasks, such as denoising[30–33] and image synthesis[34–37]. Compared with classical methods, convolutional neural network (CNN) shows remarkable performance in biological image analysis by mining subtle texture and shape changes. Since the U-Net was proposed by Ronneberger et al.[38], such type of encoder-and-decoder structure has greatly enhanced learning-based segmentation of medical images for diagnostic purposes[39]. For fluorescence images, the ability of deep learning to assist data filtering and classification has also been demonstrated[37,40]. To mitigate the complexity of cellular networks, the segmentation process is usually carried out as multiple intermediate tasks, such as nucleus detection and membrane segmentation[41,42]. For example, to harness the full power of watershed transformation, regression networks are utilized to predict a distance map, followed by different seeding procedures[43,44]. However, an integrated framework is needed to simultaneously segment membranes, trace nuclei, and identify global cell morphologies with defined cell identities over development.

Here we report an integrated pipeline, CShaper, for analyzing cell shape and constructing a 4D morphological atlas during *C. elegans* embryogenesis (Fig. 1). First, we generate a *C. elegans* transgenic strain that ubiquitously expresses both a GFP (green) and an mCherry (red) fluorescence marker in the cell nuclei and membranes, respectively, throughout embryonic development. Second, we develop a deep-learning-based method, DMapNet, to segment membranes of 17 embryos. Instead of segmenting cells as a binary classification task directly, DMapNet generates a discrete distance map from the membrane image stack. In total, it takes ~30 min for CShaper to process 3D stacks of an embryo from the 4- to 350-cell stages with an imaging interval of ~1.5 min. Outputs from nucleus-tracing tools, StarryNite and AceTree, are integrated to name the segmented regions and identify inner cavities in DMapNet (Supplementary Fig. 2). Third, we trace nuclei in another 29 embryos also with StarryNite and AceTree to allow normalization of nuclei positions and identities over time. Finally, we establish a spatiotemporal morphological atlas for *C. elegans* development from the 4- to 350-cell stages using the data of nuclei positions, cell identities, and cell boundaries, including the morphological dynamics of 656 unique cells and 479 reproducible, effective cell–cell contacts that are defined by sufficient contact area over two continuous time points. We generate such an atlas by minimizing the intrinsic and extrinsic positional variations among them through linear normalization[9]. It should be noted that 49 *C. elegans* wild-type embryos are used for three different purposes (Supplementary Data 1). While 4 of the 49 embryos (Samples 01–04) are used for network training and evaluation, 46 embryos (Samples 04–49) are used for automated lineaging by StarryNite and AceTree, which provide nuclei

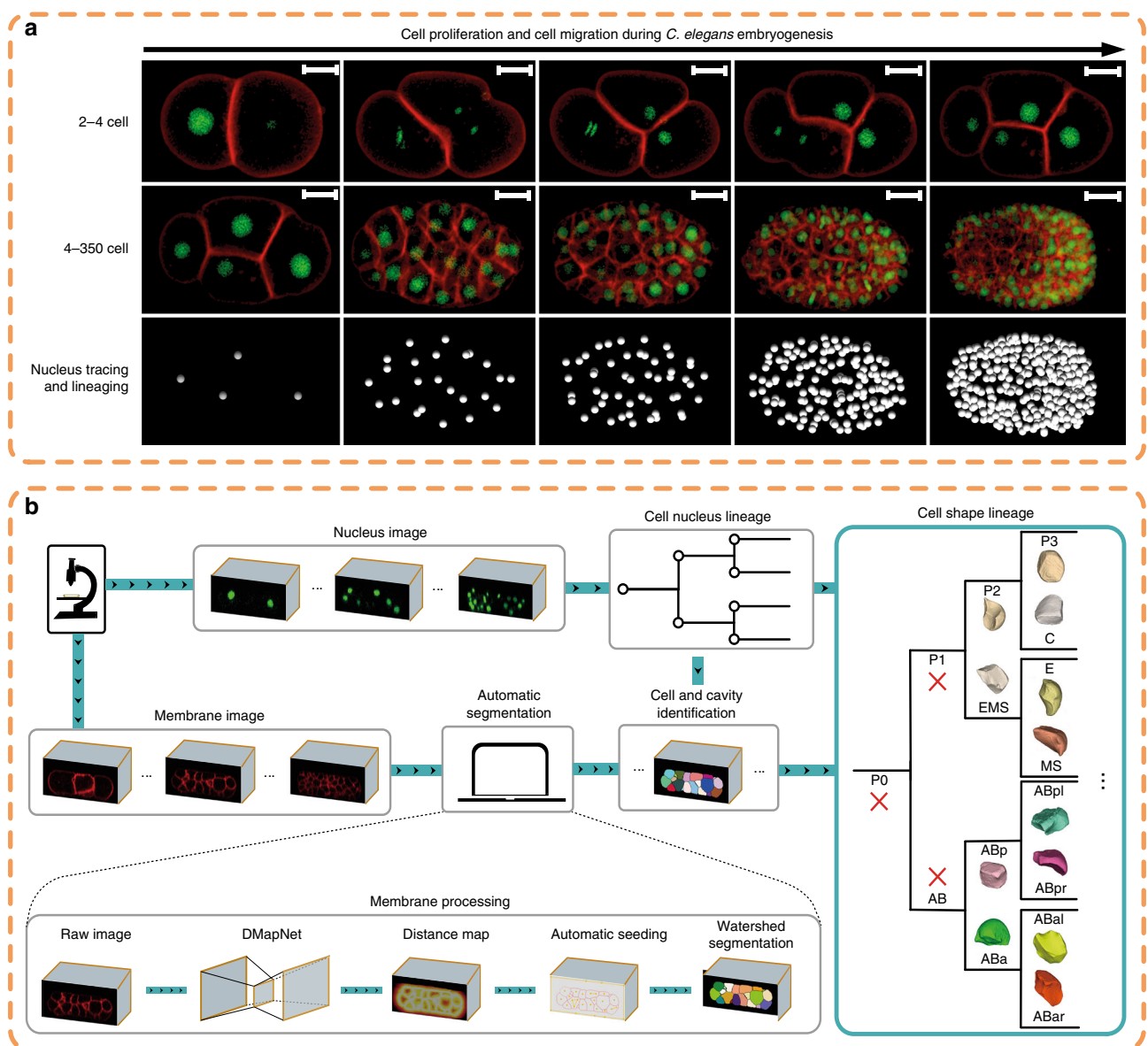

**Fig. 1 CShaper pipeline. a** 3D projections of image stacks of GFP-labeled nuclei (green) and mCherry-labeled membranes (red) at the selected time points between 2–4-cell stages (top) and 4–350-cell stages (middle). Nuclei positions determined by automated lineaging tools from the 4- to 350-cell stages are shown at the bottom. Scale bar, 10 μm. **b** Framework of CShaper. Membrane images are transformed into discrete distance maps by a distance-aware neural network, DMapNet, after which minima clustering serves as a seeding procedure for the watershed segmentation. Based on the nucleus lineage from automated lineaging tools, membrane-wrapped compartments in the segmentations with or without a nucleus are denoted as cell or cavity, respectively. Finally, time-lapse 3D cell shapes across development with defined cell identity are generated (right). Cells that do not exist during the imaging period are indicated with a red cross.

positions and identities to facilitate spatial and temporal normalization and establish a standardized atlas of the *C. elegans* embryo. Furthermore, 17 of the 46 embryos (Samples 04–20) are simultaneously labeled for both nuclei and membranes and used to generate morphological information using CShaper (see "Methods"). Evaluation of segmentation results (e.g., cell-wise overlap ratio and surface deviation) and cell morphologies established in previous experimental data (e.g., cell–cell contacts) demonstrate a robust performance of CShaper. This morphological atlas will not only facilitate the investigation of unanswered questions such as developmental variability of cell size between individual embryos, but also help revisit and reevaluate previously reported phenomena, such as active deformation of cells[45] and intercellular signaling transduction[2,14,15,46] with a higher confidence.

## Results

**Comparison of performance with existing methods.** Based on manual annotations (see "Methods"), we compared CShaper with other methods in cell segmentation, including 3DUNet[42], SingleCellDetector[43], FusionNet[47], RACE[48], and CellProfiler[49]. To allow a fair comparison, watershed algorithms were appended as a postprocessing procedure for 3DUNet and FusionNet where only binary membrane segmentation is available. However, in contrast to CShaper, the seeds were derived from the results of AceTree directly. By only replacing the last layer of DMapNet (see "Methods") with two channel filters, a variant of CShaper, termed B-CShaper, was also tested to examine the superiority of the distance-constrained learning used by CShaper to the binary classification in B-CShaper. Parameter settings and implementations for all methods are detailed in Supplementary Note 1 and

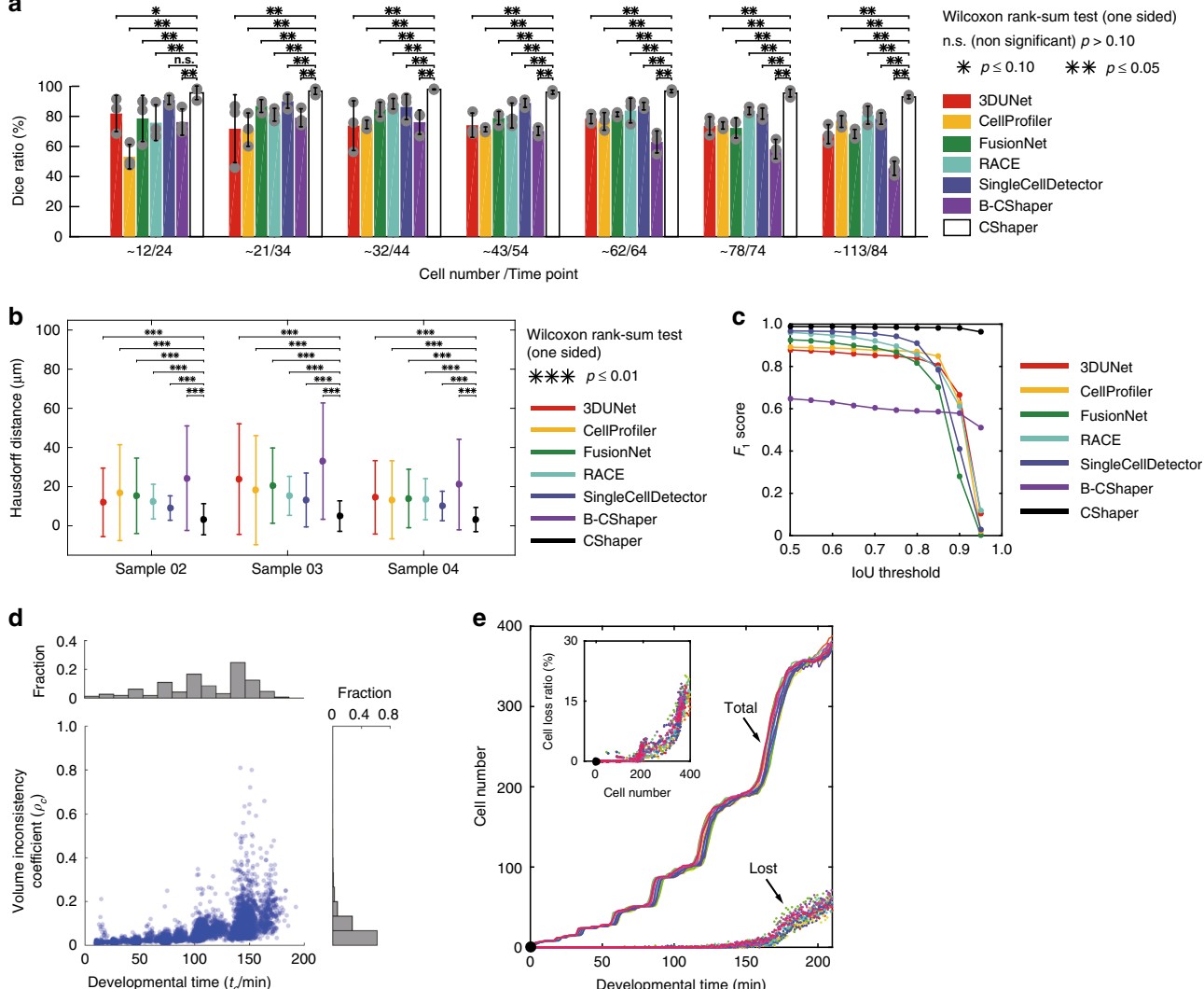

**Fig. 2 Benchmarking of segmentation results. a–c** Evaluations based on manual annotations of cells in three independent wild-type samples (02–04) with seven time points per embryo. **a** The dice ratio of the segmentations generated by 3DUNet, CellProfiler, FusionNet, RACE, SingleCellDetector, B-CShaper, and CShaper. Cell numbers are averaged at corresponding time points for each of the three embryos. Significance level is derived by one-sided Wilcoxon rank-sum test over $n = 3$ independent embryos (n.s. non significant, $p > 0.10$; *$p \leq 0.10$; **$p \leq 0.05$); error bar represents standard deviation (SD). **b** The average Hausdorff distance between the segmentation results produced by these methods and the ground truth for each sample. Significance level is derived by one-sided Wilcoxon rank-sum test over $n = 353$, 261, and 470 independent cells for Samples 02–04, respectively (***$p \leq 0.01$); error bar represents standard deviation (SD). **c** Object-level $F_1$ scores based on 1084 independent cells at different IoU thresholds. **d**, **e** Statistics describing additional 17 samples (04–20) imaged and segmented spanning the 4- to 350-cell stages. The embryos' time scales are proportionally normalized to their average. **d** Distribution of cell volume inconsistency coefficient ($\rho_c$) over time ($t_c$). **e** The number and ratio of lost cells over developmental time, where the last time point of the four-cell stage is set as the starting time point (indicated with a black point). Each color represents an individual embryo. Solid and dashed lines denote the total number of cells that were identified by nucleus tracing (total) and unsuccessfully segmented (lost), respectively. Source data are provided as a Source Data file.

Supplementary Table 1. Note that for each segmented region, labels were unified according to the ground truth based on two principles, maximizing overlapped regions and assigning a unique label to each pixel.

The dice ratio, as a pixel-level score, is widely used to measure the similarity between computational segmentation results and the ground truth. Given two areas, the dice ratio is defined as the ratio between the overlapping region and the overall area[19]. After testing embryos at different cell stages, CShaper obtained a score of 95.95 ± 2.36%, outperforming other methods, in most cases by a significant margin (Fig. 2a, $p \leq 0.05$, one-sided Wilcoxon rank-sum test). Therefore, in terms of the overlapped volume criteria, CShaper segmentation results were highly consistent with manual

annotations. Nevertheless, considering that the dice coefficient is limited when quantifying the surface misalignment in the segmentation, we simultaneously adopted the Hausdorff distance to evaluate how close the predicted surface shape approximates the reference annotation. While the Hausdorff distance represents the largest one of all distances from a voxel in one set to the closest voxel in the other set, the bidirectional Hausdorff distance was defined as:

$$d_{\mathrm{H}}(V_{\mathrm{P}}, V_{\mathrm{GT}}) = \max\left\{ \sup_{v_i \in V_{\mathrm{P}}} \inf_{v_j \in V_{\mathrm{GT}}} d(v_i, v_j),\ \sup_{v_j \in V_{\mathrm{GT}}} \inf_{v_i \in V_{\mathrm{P}}} d(v_j, v_i) \right\},$$

$$(1)$$

where $V_P$ and $V_{GT}$ are the voxel sets of segmentation and annotation for each cell, respectively. The smaller the Hausdorff distance, the better the surface approximation. As shown in Fig. 2b, CShaper had a surface deviation of $0.81 \pm 1.59$ μm, which was significantly smaller than that of any other methods ($p \le 0.01$, one-sided Wilcoxon rank-sum test). The average cellular diameter was $49.70 \pm 21.94$ μm in our evaluated samples.

The previously mentioned pixel-level scores cannot profile object-level errors, such as cell merging and splitting within an image. Therefore, we employed the segmentation $F_1$ score, proposed by Caicedo et al.[50], to benchmark the object-level performance of CShaper. First, like the dice ratio, the Intersection-over-Union (IoU) ratio $(V_P \cap V_{GT})/(V_P \cup V_{GT})$ was defined for each pair of cells. A threshold, $t$, for the IoU can divide segmentation results into three groups, TP (correctly identified cells), FN (missed cells), and FP (wrongly identified cells). With $t$ ranging from 0.50 to 0.95 by a step size $\Delta t = 0.05$, each method was evaluated repeatedly by the function $F_1 = \frac{2TP}{2TP+FN+FP}$. This $F_1$ score similarly showed that CShaper is more accurate and generates robust cell boundaries compared to previous methods (Fig. 2c). Based on the segmentation results at 0.7 IoU, we also reported the other object-level criteria, such as split, merge, precision, and recall scores, in Supplementary Table 2.

To deliver a perceptual understanding of the differences of these methods in processing the *C. elegans* images, we listed segmentation examples in Supplementary Fig. 3, corresponding to time point 84 of Sample 04 in Fig. 2a. Notably, FusionNet exhibited a severe leakage at the top of the embryo, where the single-layer membrane was too weak to be recognized due to laser power attenuation. Similarly, a loss of cells and unrecognizable boundaries were common in the segmentations produced by 3DUNet, CellProfiler, and B-CShaper. RACE and SingleCellDetector reduced the leakage to some extent by stacking slices segmented separately. However, the lack of inter-slice information introduced a distorted shape that tended to be flat at the cellular boundary. The superiority of CShaper can be ascribed to the following improvements:

(1) CShaper can restore lost membranes. In a microscopic image, cell membranes perpendicular to the focal axis are often blurry or not imaged at all. This problem is aggravated at the outer surface where only one membrane layer is available for detection. By predicting a distance map, CShaper provides an implicit shape representation in the learning process. A small local error in the distance map will affect the value of multiple points globally and further compensate for regions in low-contrast images. Essentially, the distance map can be treated as the prediction probability distribution of the voxel being the membrane, thereby making CShaper more effective for segmenting membranes with weak signal compared to the hard binary classification.

(2) The powerful seeding strategy reduces over-segmentation errors. Although watershed transformation is outstanding in instance segmentation[43,44], classical seeding methods, such as H-maximum and Mask-CNN detection, are prone to either over-segmentation or considerable computation resource. In CShaper, the weighted seeds graph allows CShaper to be easily adapted to cellular size and shape changes, permitting identification of potential cavities that are not associated with any nucleus.

**Volume inconsistency and cell loss ratio.** In order to evaluate the performance of CShaper on additional datasets when annotations are available, we quantitatively measured the volume inconsistencies and lost cells in time-lapse segmentations of 17 wild-type embryos, which were imaged from the 4- to 350-cell stages (Supplementary Data 1, Samples 04–20). CShaper segments each frame independently, without capturing typical temporal patterns; however, successively imaged cells were assumed to have temporally consistent size, although limited variance may exist when we consider specific biological dynamics such as apoptosis[1]. Therefore, we can profile the performance of CShaper by checking the inconsistency of cell volumes across many consecutive time-lapse images.

Given the cell lineages output by StarryNite and AceTree, the lifespan of each nucleus is precisely specified. Such time intervals can be used to infer the lifespans of specific cells directly except the frames at the beginning and end of each cell division, where the daughters of a dividing cell were labeled as the mother cell instead of independent cells because the two newborn nuclei still shared a membrane system. For a given cell $c$, the volume inconsistency $\rho_c$ was defined as the ratio between the standard deviation and the mean of the cell volume across its lifespan. A smaller $\rho_c$ means that the segmentation of cell $c$ has a better temporal consistency in volume, thus yielding higher segmentation performance. To differentiate errors at different developmental stages, the middle of a cell's lifespan was denoted as $t_c$, which was normalized globally and proportionally relative to the average of all cell lifespans in each of the 17 samples. The distribution of inconsistency coefficient $(t_c, \rho_c)$ indicated that most segmented cells have relatively small volume variation throughout development (Fig. 2d). Although the temporal information is not incorporated in CShaper, the resultant volumes are consistent between consecutive time points. As the increasing number of cells and the low signal-to-noise ratio impose challenges in precisely segmenting a crowd of cells, the inconsistency increases dramatically after an embryo develops up to the 200-cell stage (i.e., the start of collective divisions of AB128 cells). The $\rho_c$ may fail to capture errors when a cell is lost in too many time points within its lifespan, but such case only accounts for a small fraction in the 17 embryo samples (Supplementary Table 3). Over 80% of cells are successfully segmented with <10% of their lifespans missed. An example of the segmentation results from the 4- to 350-cell stages is provided for the visual inspection of performance (Supplementary Movie 1).

As the cell nucleus is not involved at the segmentation stage, we can also evaluate the ratio of lost cells according to pairing errors (Supplementary Note 2). It shows that prior to the 200-cell stage, the cell loss ratio is below 5% for each time point in each embryo (Fig. 2e). As the number of cells increases, the cell loss ratio increases at a much higher rate as most embryonic cells enter their ninth round of division, producing around 350 cells. However, the lost cells across the entire embryos account for only a small proportion of the total cells. During the 200–350-cell stages, the cell loss ratio is below 18% for each time point in each embryo (Fig. 2e). The data quality and reproducibility are described in detail below.

**Normalization and standardization.** Based on quality-control standards established before (see "Methods")[6,9], 46 wild-type embryos (29 with nucleus tracing alone and 17 with both nucleus tracing and membrane segmentation) imaged from the 4- to 350-cell stages were normalized and used to construct a morphological atlas of early *C. elegans* development (Supplementary Data 1). First, both cell nuclei (hereafter referred to as cell positions) and cell membranes were quantified and placed in a rectangular coordinate system, where the axes $x$, $y$, and $z$ denote anterior–posterior, left–right (L–R), and dorsal–ventral (D–V)

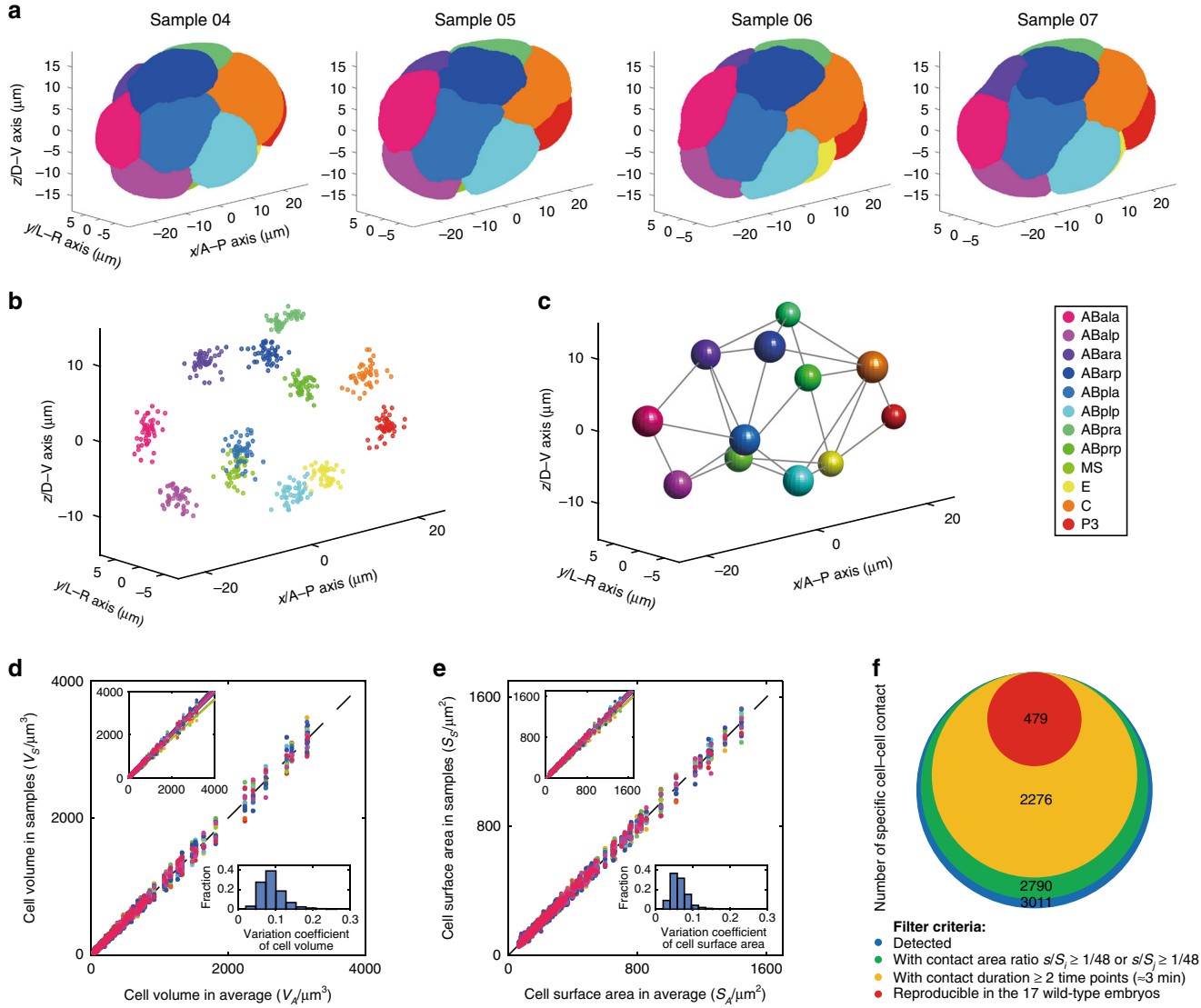

**Fig. 3 A morphological atlas during *C. elegans* embryogenesis. a–c** Different dimensions of a standardized spatiotemporal reference, illustrated using the 12-cell stage as an example. The *x*, *y*, and *z* axes represent the anterior–posterior (A–P), left–right (L–R), and dorsal–ventral (D–V) axes, respectively, and each color denotes one specific blast cell as indicated. **a** 3D representations of four wild-type embryos reconstructed with membrane label (Samples 04-07). **b** Spatial distribution of cell position (nuclei) of 46 embryos (Samples 04-49). **c** Cellular spatial deviation and reproducible cell–cell contact mapping. The sphere radius represents spatial deviation $\Delta r$ defined by the root-mean-square deviation (RMSD, $\Delta r = \sqrt{(\sum_{i=1}^{N} |r_i - \bar{r}|)/N}$, where $N$ denotes the total number of embryos and $\mathbf{r}_i$ denotes the cell position in the *i*th embryo). Each gray line represents reproducible and effective contact between cells under specific filter criteria: $s_{\text{contact}}/S_{\text{surface}} \geq 1/48$, $T_{\text{contact}} \geq 3$ min, $N_{\text{replicate}} = N_{\text{embryo}}$ (see "Methods"). **d, e** Reproducibility and variability of cell volume and cell surface area, tested using proportionally normalized data from all 322 cells with a complete lifespan (Supplementary Data 3). Each color represents an individual embryo. Top-left insets: data graphed using original data prior to normalization. Bottom-right insets: variation coefficients of each cell among the 17 embryos using the normalized data. **f** Criteria for defining effective cell–cell contact (red). The whole pool of cell–cell contacts detected and the ones filtered out through different criteria are symbolized by circles with different areas and colors. Source data are provided as a Source Data file.

axes, respectively. Second, based on the known, conserved developmental landmarks during *C. elegans* embryogenesis, i.e., the collective synchronous divisions and founder cell generations, 54 specific moments were selected to illustrate the developmental stages (Supplementary Data 2). Third, cell positions of all 46 embryos were linearly normalized to minimize their positional variation, according to a proposed computational pipeline consisting of consecutive rounds of rotation, translation, and scaling[9]. Note that the 29 samples with only nuclei positions were used to increase the sample size and help minimize the global positional variation between individual embryos. These 29 samples also provided information regarding nuclear distribution and dynamics. Finally, all embryo samples were merged and

normalized into the same framework, producing a standard dataset describing cell morphology, variation-minimized cell position and reproducible, effective cell–cell contact (Fig. 3a–c; see "Methods"). All 17 embryos with segmented cell morphologies were embodied by a unified cylindroid, approximately with a height of 18 μm, a semimajor axis of 27 μm and a semiminor axis of 18 μm (Supplementary Figs. 4–6).

**Data quality and reproducibility in successfully segmented cells.** All 17 wild-type embryos with mCherry-labeled cell membranes were processed using CShaper. To achieve high data quality and reproducibility for the atlas, we demanded that a total of 656 unique cells from different lineages, which have a complete

or partial cell-cycle length recorded, be consistently present and identified in all 17 embryos subjected to segmentation, including AB2–AB256, EMS, P2–P4, Z2 and Z3, MS1–MS32, E1–E16, C1–C16, and D1–D8 (Supplementary Data 3). Notably, 322 of these 656 cells had a complete lifespan and were successfully segmented without any frames (time points) lost in at least three embryos, while 171 of these 322 cells were successfully segmented in all 17 embryos throughout their lifespans (Supplementary Data 4 and Supplementary Note 2). The remaining 334 cells with an incomplete lifespan were segmented for their known time points. The successfully segmented cells provide quantitative information about cell morphologies, including cell shapes, volumes, surface areas, migrations, nuclei positions, and cell–cell contacts, with resolved cell identities at about 1.5-min intervals during embryogenesis.

**Cell size**. During *C. elegans* embryogenesis, cell size has been shown to be implicated in cell-cycle control, spindle formation, and differentiation[11–13]. One fascinating question in developmental biology is how an embryo maintains accuracy both spatially and temporally throughout its development. In *C. elegans*, cell-level precision has been reported for many parameters, including division timing, division orientation, gene expression, position, and migration[1,6–10,51]. However, little is known about the regulatory control of cell size and its variability over development. To this end, for both cell volume and cell surface area, we analyzed the 322 cells with a complete lifespan and normalized the data across all 17 embryos relative to their averages (Fig. 3d, e). Strikingly, the goodness of fit for both cell volume and cell surface area in each embryo is larger than 0.99, indicating a high accuracy and reproducibility in cell-size control during embryogenesis. Besides, there is an intrinsic variability of ±10% in the total embryo size (Supplementary Table 4). For each of the 322 cells, the variation coefficients of cell size among individual embryos range between around 0 and 0.2, indicating that a considerable level of variability is tolerated for both cell volume and cell surface area (Fig. 3d, e and Supplementary Table 5). Despite the relatively high variation coefficients, the size ratio between sister cells (161 pairs in total) is less variable than the overall cell size, suggesting a more precise control over the size ratio between sister daughter cells during cytokinesis than over the absolute size of each daughter. This is the case for both cell volume and cell surface area, under all the four measurements tested (i.e., average, minimum, maximum, and maximum of 99% data). For example, 99% of the individual cells have a variation coefficient of ≤0.1999 in cell volume, while 99% of the sister cell pairs have a variation coefficient of ≤0.1670 in volume ratio (Fig. 3d, e, Supplementary Fig. 7a, b, d, e, and Supplementary Table 5). Moreover, there is no significant or strong correlation between the ratio of cell volume or cell surface area and their variations for the 322 cells (161 cell pairs) among 17 embryos (Supplementary Fig. 7c, f). Availability of this standardized morphological atlas is expected to catalyze functional characterization of cellular behaviors, especially when combined with gene perturbation experiments.

**Cell–cell contact**. During metazoan development, fate induction (e.g., by Wnt[14] and Notch signaling[2,52]) and spindle formation[15,53,54] often demand direct and continuous contact between specific cells to achieve a functional interaction. For example, binding of a receptor by a ligand to achieve consequent signaling transduction[2,14,15,46,52]. Digitized embryos with detailed morphological information permit the inference of such interactions using parameters, such as contact area and duration. Given the significance of cell–cell interactions for inducing cell

fate transitions and spindle formation, we defined effective contacts between specific cells by applying three empirical criteria (see "Methods"):

(1) A contact area no <1/48 (≈2.08%) of a specific cell's surface area.
(2) A consecutive contact duration no shorter than two imaging time points (roughly 3 min).
(3) Reproducible satisfaction of the above two criteria in all 17 embryo samples.

Among the 656 cells indexed from the 4- to 350-cell stages, we selected the cells that were successfully segmented for all their time points in all 17 embryos (222 cells) for cell–cell contact identification. We detected 3011 independent pairs of cell–cell contacts, i.e., with a contact area larger than zero for at least one time point in at least one embryo, and 479 of these 3011 contact pairs were defined as effective contacts using the three above criteria (Fig. 3f and Supplementary Data 5). As these effective contacts were reproducibly observed in all embryos, it is possible that they have a function in regulating embryonic development and warrant further investigation. Several intercellular signaling events based on physical cell–cell contact have been identified experimentally as playing an important role in spindle formation, cell-fate induction, and asymmetric segregation of cytoplasmic components[2,14,15,46,52]. Here, we compared 10 well-established signaling pairs with our 479 pairs of effective cell–cell contacts (Supplementary Data 6). Most of the known contact pairs satisfy our filtering criteria, with the exception of MSapp → ABplpapp, C → ABar, and MSappp → ABplpppp. It is found that the contact between MSapp and ABplpapp was lost in 2 of the 17 embryos due to segmentation failure. In contrast, the contact between C and ABar, which is critical for Wnt signaling from the former to the latter to coordinate division orientation, was reproducibly observed in all of the 17 wild-type embryos for at least two consecutive time points (≈3 min). However, the relative contact area was smaller than the contact area cutoff in 2 out of the 17 embryos (i.e., $s_{contact}/S_{surface}$ < 1/48; Samples 04 and 17), revealing that this threshold for identifying valid cell–cell contact is expected to produce some false negatives of biologically relevant contacts. Notably, our criterion for effective contact is less stringent than that used previously (≈6.5%), which was estimated based on the contact areas of the second Notch interactions during *C. elegans* embryogenesis[2]. The contact duration between MSappp and ABplpppp lasted for only one time point (≈1.5 min) in embryo Sample 13, suggesting that the requirement of contact duration for certain cell–cell signaling may be shorter than our cutoff (two time points ≈3.0 min), which again may lead to some false negatives. This could also be due to some other redundant signaling events which compensate this interaction. Alternatively, this physical interaction may be mediated by diffusible ligands in proximity as proposed previously[55]. The false negative may be unavoidable using these specific requirements, because the actual sensitivity of intercellular signaling including both contact area and contact duration is still poorly understood. Nevertheless, these arbitrary thresholds can be readjusted in certain circumstance when the contact area and contact duration for more experimentally validated signaling events become available. For example, when a signaling event can be permitted under a less stringent physical state (i.e., relative contact area < 1/48 and contact duration < 3 min), our method and data may uncover more functional contacts in vivo.

**Cell shape**. Changes in cell shape over time reveal the dynamics of cellular mechanical properties, including, but not limited to, the passive force due to the surrounding environment, stiffness,

and adhesion[16,18]. In addition, the changing shapes of cells during development may suggest ongoing cell signal transduction, division, or migration, which are processes specifically associated with cell identity and cell fate[45,54,56]. Our membrane segmentation allows accurate delineation of cell shape dynamics throughout development with a resolution of ~1.5 min. To characterize shape changes quantitatively, we evaluated the irregularity of cell shape. For a 3D object, the surface-to-volume ratio $S/V$ is positively correlated with its irregularity. To quantify a cell's shape irregularity, we transformed both cell surface area, $S$, and cell volume, $V$, into the length dimension and calculated their ratio $\eta$ as follows:

$$\eta = \frac{\sqrt{S}}{\sqrt[3]{V}}, \qquad (2)$$

where the irregularity score $\eta$ only depends on the shape of an object but not on its volume or surface area, and will reach its minimal value, $2^{\frac{1}{3}} \times 3^{\frac{1}{3}} \times \pi^{\frac{1}{6}} \approx 2.1991$, in an ideal sphere, as calculated by the well-known formulae of sphere volume ($V = \frac{4}{3}\pi R^3$) and surface area ($S = 4\pi R^2$). The ratio $\eta$ increases with the number of tiny spikes on the surface as shown by the five Platonic polyhedra in Fig. 4a[57]. We calculated the dynamics of $\eta$ for all 322 cells with a complete lifespan (Fig. 4b). The irregularity $\eta$ ranges from 2.3479 to 2.8060 in the cells examined across their lifespans, while their temporal averages range from 2.4011 to 2.5934, suggesting that the cells are mostly deformed in a severe level similar to those in octahedron, cube, and tetrahedron, and are not as round as a perfect sphere, icosahedron, or dodecahedron (Fig. 4a, b and Supplementary Data 7). As with the division timing, the progenies of the P1 cell are less regularly shaped than those of the AB cell until roughly the 200-cell stage (Fig. 4b and Supplementary Fig. 8). Regarding the irregularity scores averaged over the cells' lifespan, the top 10% cells with the most irregular shapes consist of 11 AB progenies and 22 P1 progenies, though the total number of AB progeny is about twice of that of P1 (Supplementary Data 7). In terms of the last generation of cells examined, the AB128 cells have an irregularity score of 2.4437 ± 0.0174, while the E8 cells reach a higher score of 2.5206 ± 0.0228, indicating more irregularity in the shape of E8 cells compared to AB128 cells. The exceptionally high score seen for E8 cells seems to be due to numerous spikes or wrinkles on the cells' surfaces (Supplementary Fig. 9).

Further characterization of the correlation between shape irregularity and cell fate sheds light on how cell shape is coupled with fate differentiation during morphogenesis and organogenesis. For example, some highly irregular cells seem to play a leading role in driving morphogenesis. The two most irregular cells, MS ($\eta \approx 2.5934$) and ABpl ($\eta \approx 2.5913$), are among the first eight cells generated in the zygote after the third round of divisions, and contact each other continuously (Fig. 4b). Both cells show a severe deformation with multiple sharp humps on the edges, particularly in the middle of their lifespans, while other cells from the same generation resemble an ellipsoid, with a much smoother surface (e.g., ABal and P3, Fig. 4c). Consistent with this, previous studies have also noted dramatic shape changes in ABpl and MS, which are proposed to promote L–R patterning during early *C. elegans* morphogenesis[45]. The ABpl cell is in the center of embryo and actively drives cell rearrangement. ABpl also migrates in an anterior-ventral direction, with the longest migration distance recorded over early *C. elegans* development[9], suggesting a mechanistic link between cell migration and cell shape (Fig. 5a).

To investigate the reproducibility and variability of cell morphology across development, we compared the changes in ABp, ABpl, and ABpr morphology throughout their lifespans in a subset of six embryos (Samples 08, 10, 12, 14, 15, and 19)

(Fig. 5a). According to our 4D morphological data, the division angle of ABp deviated from the D–V axis varies among individuals. In the embryo Samples 08, 12, 14, and 15, the sister cells ABpl and ABpr are oriented roughly parallel to the AP–LR plane, without apparent bias toward the D–V axis. However, the orientation of ABp division in the embryo Samples 10 and 19 is tilted toward the D–V axis, possibly affected by the upcoming division of the EMS cell. Nevertheless, in all embryos, ABpl moves in the anteroventral direction away from its sister cell once it becomes an independent cell. These observations suggest that ABpl's migration in the ventral direction is independent of its mother cell's division orientation, despite the cell division orientation being strictly controlled by both genetic and mechanical mechanisms to establish the L–R axis in the embryo[9,45,54]. Notably, although ABpl and ABpr show reproducible contact between each other during early stage of their cell cycle, their contact becomes dispensable during later stage (Fig. 5a). For example, only 8 of 17 embryos show a considerable contact between ABpl and ABpr (Fig. 5b). These results demonstrate that although *C. elegans* embryogenesis is well known for its accurate spatiotemporal regulation of cell behavior during development, substantial variability of cellular morphology and behavior is tolerated. Through systematic quantification of cell shape, volume, surface, and contact variability, it may be possible to predict essential regulatory activities that manifest as stereotyped morphological dynamics during development. For example, unlike the ABpr cell, ABpl appears as a severely deformed polyhedron with rough surface patches and sharp humps. These humps, consistent with morphological features of cellular membranes including lamellipodia, protrusion, and filopodia, may play a pivotal role in ABpl's active migration toward the anteroventral direction[45]. However, the shapes of both ABpl and ABpr eventually become isotropic polyhedrons with smooth surfaces prior to their divisions, suggesting that biophysical properties such as stiffness, surface tension, and active motility cease to significantly influence cell shape immediately prior to cell division. In principle, cell morphology in 3D space can be projected onto spherical coordinates to form a 2D distribution. This strategy can allow a more quantitative and systematic profiling of our data, rather than characterizing cell shape with 1D data. These observations, based on ABp, ABpl, and ABpr, can be readily applied to any other cells across the 4– to 350-cell stages in *C. elegans*, with complete and reproducible segmented data.

## Discussion

Cell morphology plays an important role in various biological processes. Here we established a pipeline, CShaper, for analyzing spatiotemporal morphological features of the *C. elegans* embryo at cellular resolution with ~1.5-min intervals and resolved cell identities during embryogenesis. CShaper benefits from a well-defined distance learning model DMapNet. By learning to capture multiple discrete distances, DMapNet extracts the membrane contour while considering shape information, rather than just intensity features. The performance of CShaper was examined at both pixel and object levels. Based on the segmentation results, we integrated the data from 46 embryos and generated a quantitative morphological atlas of the developing *C. elegans* embryo from the 4- to 350-cell stages, including cell shape, volume, surface area, migration, cell nucleus position and identity, and effective cell–cell contact throughout development. We presented reproducible morphological dynamics for 322 cells throughout their lifespans, and additional 334 cells for their partial lifespans. Furthermore, based on contact area and duration, we generated 479 effective contacts between specific cell pairs, which may

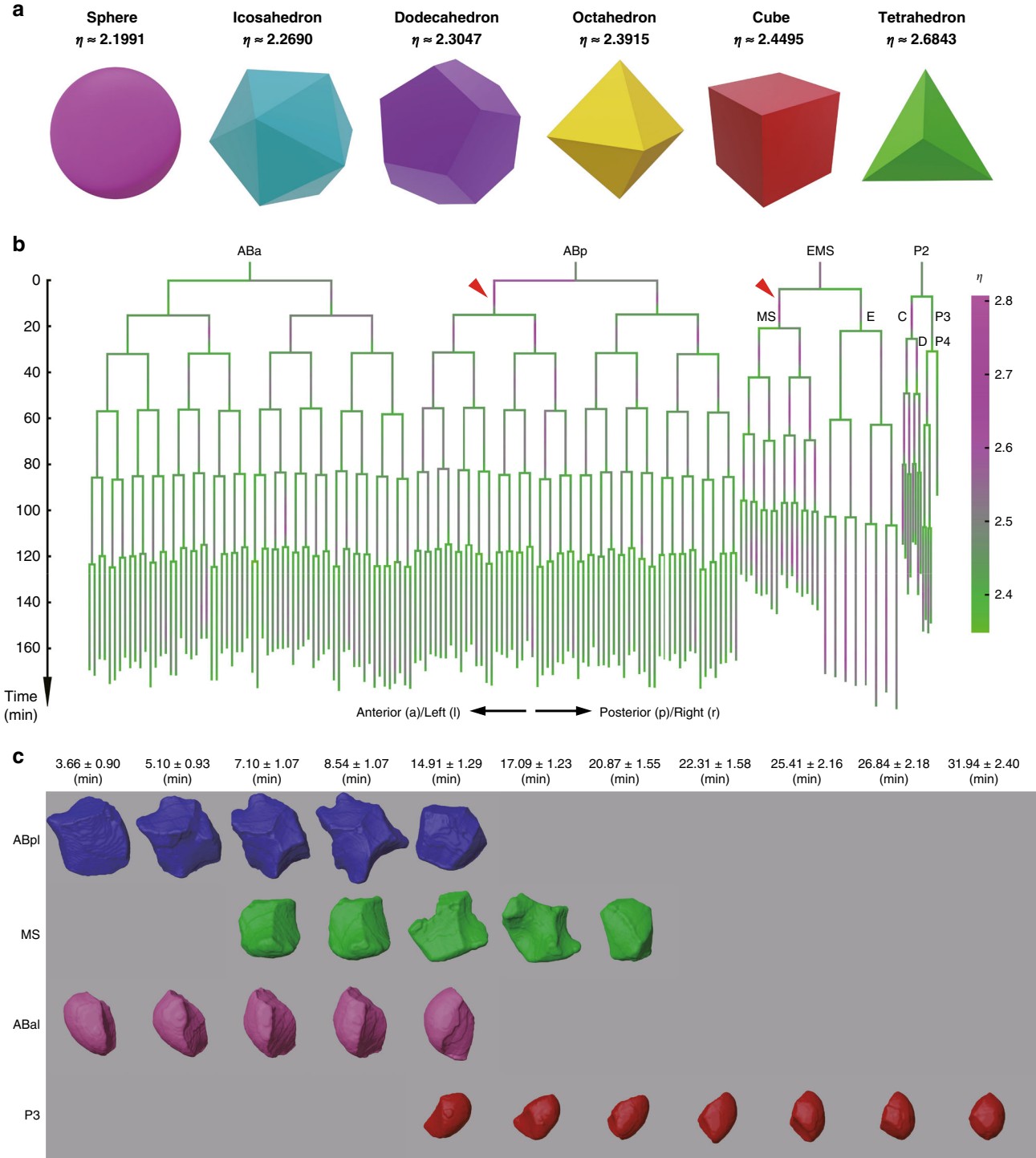

**Fig. 4 Cell shape irregularity during *C. elegans* embryogenesis. a** Dimensionless irregularity scores, $\eta$, for a sphere, icosahedron, dodecahedron, octahedron, cube, and tetrahedron. **b** Distribution of cell shape irregularity over development shown in a cell lineage tree, with the two most irregular cells, ABpl and MS, indicated by red triangle. The color scheme denoting the level of shape irregularity is shown on the right. Source data are provided as a Source Data file. **c** Time-lapse 3D cell shape dynamics of ABpl (blue), MS (green), ABal (pink), and P3 (red) derived from embryo Sample 05. Horizontal and vertical axes represent anterior–posterior (A–P) and dorsal–ventral (D–V) axes, respectively. Dynamics of 3D shape is shown over developmental timings (mean ± standard deviation) indicated at the top (Supplementary Data 2).

functionally inform cell–cell interactions such as intercellular signaling and tethering.

The robust performance of CShaper is further supported by existing reports based on experimental data gathered during *C. elegans* embryogenesis. For example, using manually curated cell surfaces, Arata et al.[11] reported a power law relationship between

cell-cycle duration and cell volume in early *C. elegans* development, in which AB and MS cells adopt the same power exponent ($\approx -0.27$), while C and P cells share a smaller value ($\approx -0.41$). Using a log–log scale coordinate system, we performed linear fitting of our data for cell-cycle duration and cell volume systematically, demonstrating that the power exponent of AB and

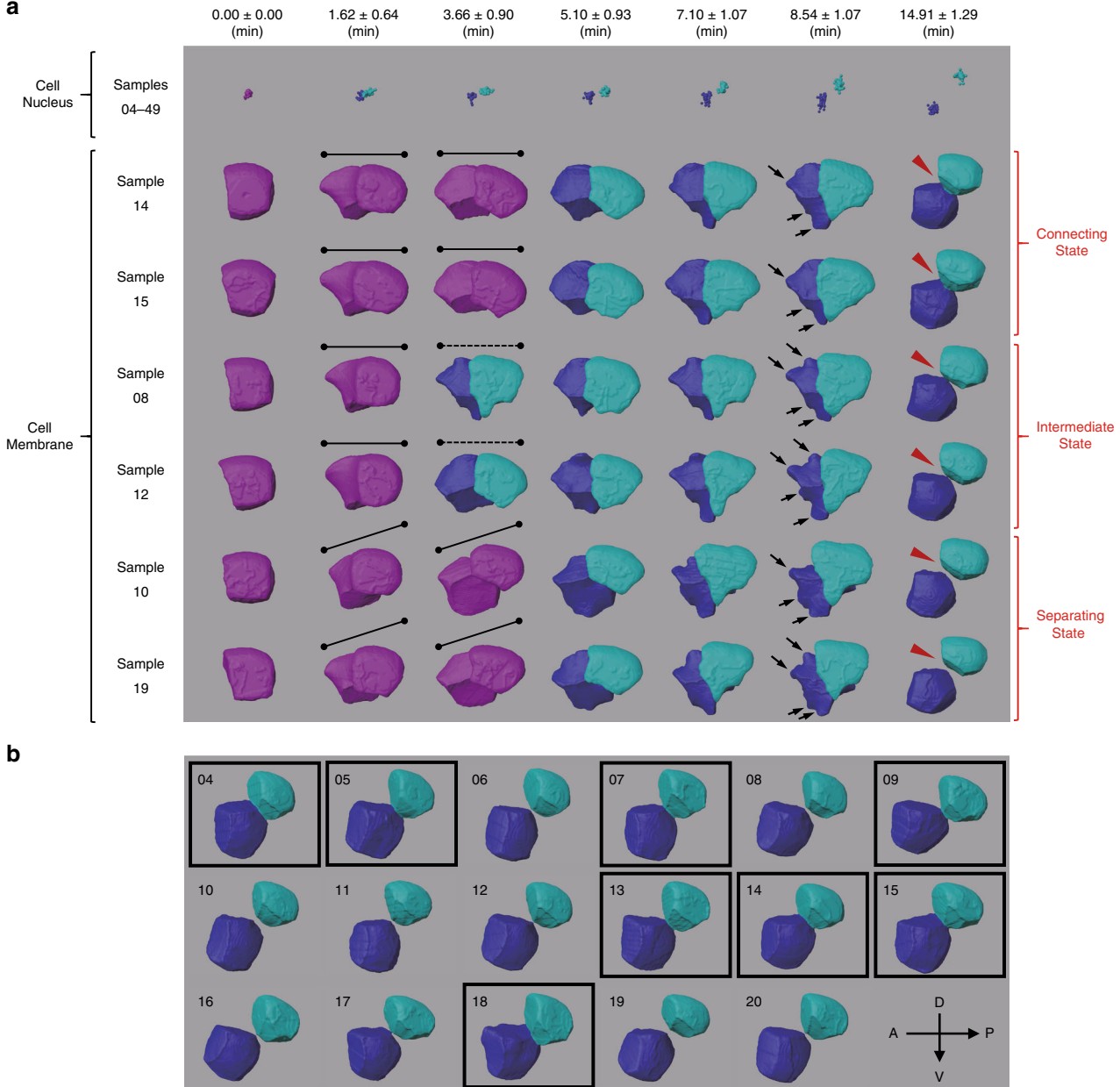

**Fig. 5 Morphological dynamics at single-cell resolution.** Time-lapse 3D cell shapes during *C. elegans* embryogenesis for ABp (purple) and its daughter cells, ABpl (blue) and ABpr (cyan). Horizontal and vertical axes represent anterior–posterior (A–P) and dorsal–ventral (D–V) axes, respectively. **a** Morphological dynamics are shown as in Fig. 4c for the time points immediately before and after AB2 divisions (columns 1–2), before and after EMS division (columns 3–4), before and after P2 division (columns 5–6) and before AB4 divisions (column 7) (Supplementary Data 2). The first row shows the cell position (nuclei) distributions obtained from all 46 wild-type embryo samples (04–49). The second to seventh rows show reconstructed cell morphologies from embryo samples segmented with membrane markers. Embryo Samples 08, 10, 12, 14, 15, and 19 were selected to demonstrate the morphological dynamics of ABp and its daughter cells. Note that in the second and third columns the division orientation of ABp is marked by solid and dashed lines for unseparated and separated daughter cells, respectively. In the sixth column, sharp humps on the ABpl's surface are highlighted by black arrows. In the last column, ABpl and ABpr can be observed in a connecting state, intermediate state or separating state, exemplified by Samples 14 and 15, Samples 08 and 12, Samples 10 and 19, respectively. **b** Position and morphology of ABpl (blue) and ABpr (cyan) at the time point prior to AB4 divisions. The embryo samples with ABpl-ABpr cell pairs in a connecting state are highlighted by black squares.

MS progeny reliably recapitulates what was proposed before (−0.293) (Supplementary Fig. 10). The power exponent of C and P progeny is substantially smaller than that in AB and MS progeny (−0.363). This allows us to classify these four lineages into two groups based on cell-cycle control, i.e., as being either highly or moderately coupled with cell volume, as proposed previously[11]. It is worth noting that in contrast to manual annotations or theoretical approximations, our membrane-calibrated

data set was generated automatically with defined cell identity, which is expected to be more accurate and systematic, allowing upscaling to both wild-type and mutant samples.

The extracellular cavity that cannot be assigned to any nucleus or cell may have specific biological meaning (Supplementary Fig. 2). Prior to gastrulation at the 26-cell stage, a blastocoel is gradually formed at the center of embryos through cooperation of cell polarization and cell adhesion[58]. This blastocoel is composed

of a set of subtle inner cavities and provides a space for the upcoming ingression of the intestine precursors, Ea and Ep (i.e., gastrulation). However, systematic characterization of such cavities is still lacking. Based on our segmentation results, a blastocoel consisting of three discrete cavities is shown in Supplementary Fig. 11. As these cavities can be systematically captured by CShaper, at least partially, our segmentation pipeline enables quantitative study of such intercellular spaces and their underlying mechanisms and functions.

To explore the capability of CShaper in segmenting images from species other than *C. elegans*, we applied CShaper to plant tissue images generated previously[59]. Willis et al. segmented *Arabidopsis thaliana* stem cells with MARS[60], which enabled a satisfactory discrimination of the inner parts of the tissue by fusing multi-angle acquisitions. Unlike MARS, CShaper processes the stem cells in a more technically challenging manner, i.e., segmenting membrane stacks from a single imaging direction, without a fusion stage. Due to the substantial morphological differences existing between animal and plant cells, we retrained DMapNet with two segmentation results from MARS. Similar to CShaper's framework except the final cell identification stage, the retrained model was then utilized to process test images. A segmented region was filtered out when its size deviated from the average volume too much (by ≥80%). Finally, we compared our cell-level segmentation results with those from MARS (Supplementary Fig. 12). When the image quality was high, in the shallow layers, both MARS and CShaper rendered comparable partitions. CShaper achieves a superior performance over MARS in the inner parts of the tissue, where the light intensity degrades significantly. Despite being trained with defective references (Supplementary Fig. 12b, d), learning shape features enables CShaper to discriminate blurred membranes. These results show that CShaper is readily adapted to analyze image data from samples other than those of *C. elegans*.

Although CShaper performs relatively well across the 4– to 350-cell stages of *C. elegans* embryogenesis, the accuracy of segmentation deteriorates at a much faster pace during later developmental stages. First, as the cell shape changes continuously over time, the temporal features between consecutive frames can be integrated to improve the segmentation performance. Long short-term memory (LSTM), originally designed for the natural language process, is an obvious candidate to capture temporal features across time[20]. However, CShaper does not adopt an LSTM-based model, such as ConvLSTM[61], due to the considerable computational resources involved in 3D convolution. We also observed that the segmentation errors of CShaper are concentrated at the top of the imaged stack, where the membrane signal intensity decreases significantly due to laser attenuation along the imaging direction during image acquisition. Within the framework of CShaper, potential strategies could be used to normalize the image quality of the top half of the embryo based on the bottom half. For example, Generative Adversarial Networks can be used to transform low-quality images into those with a higher resolution[31,34]. Furthermore, improved image quality or a novel segmentation algorithm may allow a better performance especially at later stages when cellular size substantially decreases and cells become more crowded.

Based on CShaper, 17 wild-type embryos with segmented cell membranes are provided as a resource for further studies. However, a user-friendly visualization tool is still lacking, which is critical, especially for biologists, to deliver perceptual cellular morphological features in both longitudinal and transversal directions. MorphoNet[62] may serve as an alternative tool for displaying our morphological data, though it requires considerable adaptions. A well-designed platform that allows interactive navigation of cell morphologies and contacts over development is required in the future.

## Methods

**C. elegans strain**. All animals were maintained on NGM plates seeded with OP50 at room temperature. Using Gibson Assembly, the construct P*his-72*::PH (PLC1delta1)::mCherry::*pie-1*-3′UTR was made and cloned into a *miniMos* vector[63] for transgenesis. The *His-72* promoter and *pie-1* UTR were used to achieve broad expression in both the soma and germline. A membrane labeling strain, ZZY0637, carrying a single copy of this transgene, was generated using the *miniMos* technique[63]. It was crossed with the nucleus labeling strain, RW10029, which ubiquitously expresses a fusion between histone (HIS-72) and GFP, enabling automated tracing and identification of nuclei[64]. Both the nucleus and membrane markers were rendered homozygous in the resulting strain, ZZY0655, before automated lineaging and membrane segmentation. The genotypes of the strains used in this study are listed in Supplementary Data 8.

**Image acquisition**. The imaging method of the strains expressing nucleus label only (RW10112 and RW10348) was the same as Ho et al.[6]. A similar method for the imaging of the strain expressing both nucleus and membrane labels (ZZY0655) was modified in this study as follows. One- to four-celled embryos were dissected from the adult worms. They were mounted for imaging using 1% methylcellulose in Boyd's buffer with 20 μm Polybead® microspheres (Polysciences, Inc.)[6,21,65]. Imaging was performed with an inverted Leica SP5 and SP8 confocal microscope equipped with two hybrid detectors at a constant ambient temperature of 21 °C. Images were consecutively collected for both GFP and mCherry channels using a water immersion objective. By using a resonance scanner, both channels were imaged with scanning speed of 8000 Hz with a frame size of $712 \times 512$ pixels per channel. The excitation laser beams used for GFP and mCherry are 488 nm (SP5 and SP8) and 594 nm (SP5) or 552 nm (SP8), respectively. Histone::GFP was used as a lineaging marker for cell tracing later, whereas PH2::mCherry was used as a membrane marker. Fluorescence images from 68 (SP5) or 70 (SP8) Z-steps were collected consecutively for three embryos per imaging session with a Z-resolution of 0.42 μm (SP5) or 0.43 μm (SP8) from top to bottom of the embryo for every time point, which was at ~1.5-min interval. Images were continuously collected for at least 130 time points during which the cell count would reach over 350 in a wild-type embryo. The entire imaging duration was divided into four time blocks by time point, that is, 1–60, 61–130, 131–200, and 201–240. Z axis compensation was 0.4–4% for the 488 nm laser and 19–95% for the 594 nm laser in SP5, whereas 0.1–0.3% for the 488 nm laser (SP8) and 2–10% for the 552 nm laser (SP8). The pinhole sizes for the four blocks were 2.3, 2, 1.6, and 1.3 AU, respectively. Prior to image analysis, all images were subjected to deconvolution followed by resizing into isotropic volume images with a resolution of either 0.22 μm (for training or evaluation) or 0.25 μm (for morphological atlas generation).

**Nucleus tracing and lineaging**. The nuclei images were segmented and identified using StarryNite and visualized using AceTree[21,22,64]. The lineaging errors were manually corrected up to the 350-cell stage. The data quality was confirmed using the quality-control standards designed by Guan et al.[9]. First, all the embryos must start to be imaged before AB2 divisions so that information of the four-cell stage can be obtained (i.e., ABa, ABp, EMS, and P2), which is essential for spatial normalization of different embryos. Second, the full lifespans of AB4–AB128, MS1–MS16, E1–E8, C1–C8, D1–D4, P3, and P4 cells have to be recorded. Third, their descendants, namely AB256, MS32, E16, C16, D8, Z2, and Z3 cells, have to be present for at least one time point (Supplementary Data 3). Finally, the nucleus information, including position and name, was output in a separate file to be used for cell-membrane segmentation.

**Manual annotation of cell**. Here a new data set was annotated and used to train the DMapNet and benchmark CShaper against existing methods. As only 2D slices can be shown on a computer screen, it is nontrivial to fully annotate volumetric data. Therefore, a gold standard data set was generated in a semiautomatic manner, in which segmentation errors from software were manually corrected by experts. The membrane stack was first pre-segmented by a traditional method for 3D membrane morphological segmentation (3DMMS)[66], and then the output was checked by two experts with an interactive tool for semiautomatic segmentation of multi-modality biomedical images (ITK-SNAP)[67] slice-by-slice. To aid manual examination of cavities formed among the neighboring cells, nuclei images were incorporated alongside the membrane-based images. Most annotated embryos had fewer than 100 cells to prevent the deterioration of annotation accuracy with image quality and subsequent segmentation errors introduced by 3DMMS. The annotations are composed of cell-wise regions, which can be easily transformed into membrane masks through morphological operations. For training, 54 volumetric stacks with an average of 65 cells in each were annotated. Another 21 stacks with an average of 52 cells in each were also annotated in parallel for independent evaluation. A full summary of annotation data sets is provided in Supplementary Table 6.

**Distance-constrained learning**. During long-duration time-lapse imaging, it is desirable to collect each pixel with a sufficient number of photons, but the imaging frequency has to be limited to keep the embryo alive. Segmenting the embryo with

low image quality is a challenging task. To solve this problem, a distance-aware network, DMapNet, is proposed to learn the cell shapes implicitly. DMapNet is able to discriminate weak membranes, especially in the periphery where only a single layer of membrane exists.

Although the distance map was previously discussed in similar works[43,44], different strategies are used in CShaper to facilitate the learning and postprocessing stages (Supplementary Figs. 13 and 14). Given the input image $\mathbf{I}$, we define $\mathbf{\Phi}_i(\mathbf{I})$ as the corresponding ground truth of the binary membrane at pixel $i$, where foreground membrane and background pixels have values 1 and 0, respectively. With the membrane mask forming a single-pixel surface, the distance map $\mathbf{\mathcal{M}}$ is formulated as:

$$\mathbf{\mathcal{M}}_i = \begin{cases} \min_{\mathbf{\Phi}_j=1}\left\{(x-x_0)^2+(y-y_0)^2+(z-z_0)^2\right\}, & \mathbf{\Phi}_i = 0 \\ 0, & \mathbf{\Phi}_i = 1 \end{cases}, \quad (3)$$

$$\mathbf{\mathcal{M}} = \tau(\max\{\mathbf{\mathcal{M}}\} - \mathbf{\mathcal{M}}, d), \quad (4)$$

where $x, y, z$ and $x_0, y_0, z_0$ represent the coordinates of pixels $i$ and $j$, respectively. In Eq. (4), we reverse the distance map to keep it monotonically decreasing from the membrane to the background. The background here includes both cell interiors and external embryo background. Subsequently, a truncation function $\tau(*, d)$ sets values above $d$ to $d$ or otherwise retains the value. Due to the lack of distinctive features among far-away voxels, $d$ is chosen such that it constitutes a smooth transformation from the foreground membrane to the background. By predicting $\mathbf{\mathcal{M}}$, DMapNet outputs the unnormalized probability of the voxel being the membrane. As emphasized by Peter et al.[68], $\mathbf{\mathcal{M}}$ is further nonlinearly discretized into $\mathbf{\mathcal{M}}^{\mathrm{d}} \subset \{0, 1, \ldots, K\}^3$, the learning target, with smaller intervals around the membrane mask. The cross-entropy loss used to evaluate the learning progress is defined as:

$$l = \sum_{i=1}^{N}\sum_{k=1}^{K}\xi_k\omega_{i,k}\left(\mathbf{\mathcal{M}}_{i,k}^{\mathrm{d}}\log\mathbf{P}_{i,k} + \left(1-\mathbf{\mathcal{M}}_{i,k}^{\mathrm{d}}\right)\log\left(1-\mathbf{P}_{i,k}\right)\right), \quad (5)$$

where $\mathbf{\mathcal{M}}_{i,k}^{\mathrm{d}}$ is the $k$th element of the one-hot target vector at pixel $i$, and $\mathbf{P}_{i,k}$ is the counterpart in the output of DMapNet. $N$ and $K$ are the numbers of pixels and distance intervals, respectively. The importance of different classes is adjusted by the fixed weighting term $\xi_k$, which inclines to classes near the membrane. We also incorporate interclass relationships into the loss through an interclass weighting term, $\omega_{i,k}$. Compared with $\xi_k$, $\omega_{i,k}$ dynamically changes depending on different predicted classes. This strategy is derived from the assumption that in ordered class prediction, one class closer to the ground truth is supposed to have a larger predicted probability. For example, for class $k=1$, a higher penalty should be imposed to a predicted class $k'=15$ than that of $k'=2$. Therefore, if the $K$th interval denotes the center mask of the membrane, interclass weight $\omega_{i,k}$ is calculated with:

$$\omega_{i,k} = \exp\left(\frac{|k-\mathbf{\mathcal{M}}_i^{\mathrm{d}}|}{K}\right), \quad (6)$$

where $\mathbf{\mathcal{M}}_i^{\mathrm{d}}$ is the ground truth class at pixel $i$.

**Network structure.** Although a 3D deep network has the advantage of capturing holistic features, the lack of computational resources and training data may limit the application of such model. Consequently, a pseudo-3D data flow is utilized throughout the network of DMapNet. In confocal imaging, a 3D stack is acquired by optical sectioning of embryos in the depth direction. Considering the thickness of the membrane, as well as the elongated light volume emitted by a single fluorescent molecule, only 24 consecutive slices are cropped as the input to DMapNet. It follows the structure of the U-Net with high-level abstraction information extracted by a down-sampling path and low-level features assembled by an up-sampling path. To ensure efficient gradient propagation, multiple residual blocks are leveraged at different down-sampling levels. While a $3 \times 3 \times 3$ kernel can be decomposed into $3 \times 3 \times 1$ and $1 \times 1 \times 3$ kernels, the residual block only includes the $3 \times 3 \times 1$ kernel in addition to the group normalization and Parametric Rectified Linear Unit layers. Before the max pooling, the $1 \times 1 \times 3$ kernel is used to fuse the features of multiple channels. Dilation convolution is added to enlarge the receptive field. To aid the higher layers retain the raw image information, the input is scaled down and concatenated with corresponding high-level feature maps, which also boosts the performance in segmenting cells of different sizes[40]. In the up-sampling stage, all linearly up-sampled features are convoluted with the $3 \times 3 \times 1$ kernel before being concatenated together. The class-wise probability $\mathbf{P} : D \times W \times H \times K \in [0, 1]$ is obtained by another convolution of the assembled features. Thereby, the distance map $\mathbf{\Psi} : D \times W \times H \subset \{0, 1, \ldots, K\}$ can be easily derived from $\mathbf{\Psi}_i = \arg\max_k \mathbf{P}_{i,k}$.

An overview of the DMapNet architecture is shown in Supplementary Fig. 15. Because eight boundary slices at the apex and base are excluded in the loss function, the dimensions of input and output are $24 \times 128 \times 128$ and $16 \times 128 \times 128$, respectively. DMapNet was implemented with TensorFlow and Python. Inputs were randomly cropped from 54 volumetric stacks of the size of $134 \times 205 \times 285$. Adam optimization with an initial learning rate of $5 \times 10^{-4}$ was used to update parameters. By setting the batch size to 2, we trained the model for 5000 epochs on one NVIDIA 2080Ti GPU. Both the data set and source code are available publicly.

**Watershed segmentation with automatic seeding.** Watershed segmentation is well suited for separating individual cells based on the distance map $\mathbf{\Psi}$. Although promising, the application of watershed transformation to the map suffers from over-segmentation, where a single cell is split into multiple regions. Here, we propose an automatic seeding procedure to facilitate the cellular segmentation by detecting appropriate seeds from the membrane mask.

The $K$th class in $\mathbf{\Psi}$ is regarded as the membrane mask $\mathbf{\Phi}^{\mathrm{p}}$. By selecting the background as the target voxel, Euclidean distance transformation is applied to $\mathbf{\Phi}^{\mathrm{p}}$, yielding $\mathbf{\mathcal{M}}^{\mathrm{p}}$. All local H-minima in $\mathbf{\mathcal{M}}^{\mathrm{p}}$ are denoted as $S = \{s_i\}_{i=1,\ldots,L}$, where $L$ is the number of local minima. A weighted graph $G$ is constructed to cluster $s_i$'s that belong to one cell or background. Edges $E = \{E_1, E_2\}$ in $G$ come from two sources: one is the Delaunay triangulation on $S$, noted as $E_1$, and the other is the edges, $E_2$, among all local minima locate on the boundary of the volume. The weight of edge $e_{ij}$ is defined as:

$$W\left(e_{ij}\right) = \begin{cases} \sum_{(x,y,z)\in e_{ij}}\mathbf{\mathcal{M}}^{\mathrm{p}}(x,y,z), & e_{ij} \in E_1 \\ 0, & e_{ij} \in E_2 \end{cases}, \quad (7)$$

where $(x, y, z)\in e_{ij}$ represents all points on the edge $e_{ij}$. One edge is removed from $E$ if the corresponding weight is greater than the OTSU[69] threshold on $W$. Consequently, vertexes $S$ were clustered based on their connectivity. As opposed to inspecting each minimum, we treated each cluster, possibly including multiple minima, as one seed. This group-seeded watershed transformation on $\mathbf{\mathcal{M}}^{\mathrm{p}}$ reduces under- or over-segmentation errors. A schematic description is also provided (Supplementary Fig. 16).

**Cell tracing and identification.** StarryNite and AceTree were used to automatically trace and assign identity of each nucleus by outputting nuclei positions and names. In CShaper, we leveraged these tools to name the segmentations described above. Generally, if one partition was associated with only a single nucleus, then the cell was named after the nucleus directly. However, at the beginning of cell division, two nuclei may coexist within one cell (enwrapped in the same membrane) during anaphase. In this case, the segmented region was named after the mother cell rather than the daughter cells. CShaper also defined a cavity inside an embryo when a partition was empty with no nucleus inside. As the nucleus was not involved in the membrane segmentation directly, we can not only identify lost cells based on the mismatch between nucleus and partition regions (Supplementary Note 2) but also evaluate the segmentation performance at object level (see "Results").

**Standardization of embryo samples.** After linear normalization of the 46 embryos (Samples 04–49) as per the previously proposed pipeline, which consists of consecutive rounds of rotation (60 cycles), translation (60 cycles), and scaling (30 cycles) in $x$, $y$, and $z$ axes to minimize the global positional variation between embryo samples[9], four operations were subsequently carried out to establish a standard morphological atlas with normalized embryo size and orientation. First, a translation in the $yz$ plane and rotation around the $x$ axis was performed sequentially on the 17 embryos that expressed the membrane marker (Samples 04–20), which ensured that the focal planes of the first and last confocal images were parallel to the $xz$ plane and distributed symmetrically on both sides of the $xz$ plane (Supplementary Fig. 4). Second, a translation in the $xz$ plane and rotation around the $y$ axis were performed sequentially on the same embryos to keep their projection on the $xz$ plane embedded by a centralized ellipse with the minimum area (Supplementary Fig. 5). Third, all 17 embryos were rescaled to their average size in the three orthogonal directions (Supplementary Fig. 6). Finally, the remaining 29 embryos labeled with only the nucleus marker (Samples 21–49) were linearly normalized to the average cell positions of the 17 embryos using the same loop algorithm composed of rotation, translation, and scaling (Fig. 3b)[9].

**Definition of effective cell–cell contact.** The following empirical criteria were used to establish effective contact between specific cells with potential biological relevance:

1. Contact area: a contact area is no <1/48 of a cell's surface area. This area threshold is expected to be large enough for functional intercellular communication based on theoretical modeling. It is well known that each sphere is surrounded by 12 neighbors in a close-packed structure of equal-sized spheres, which in theory has the highest space occupancy and system stability[70]. In the *C. elegans* embryo, the radius ratio between neighbor cells can reach up to 3:1 (Supplementary Fig. 17a). Thus, based on the hexagonal close-packed structure, we estimated the cell–cell contact area threshold by simulating how many cells with a radius ratio of up to 1/3 can be accommodated within space formed by a unit cell with a radius of 1 (Supplementary Fig. 17b–d). As a uniform neighbor cell can be replaced by, at most, four smaller cells with a radius ratio of 1/3 (Supplementary Fig. 17e), the relative contact threshold was set as $1/12 \times 1/4 = 1/48$ (Supplementary Note 3).

2. Contact duration: a contact duration is no shorter than 3 min, i.e., two consecutive time points. This threshold was previously found to be satisfied by all the cell pairs with known Notch signaling in *C. elegans*[2].

3. Reproducibility: a contact is reproducible in all 17 embryos. As we focused on cell–cell contact necessary for normal development, reproducible contacts found in all samples were assumed to have the highest possibility of being functional. This requirement (100%) is higher than that used by Chen et al.[2] (95%) because the contact relationship obtained based on the membrane morphology is expected to be more reliable than that inferred from the nucleus position.

**Reporting summary**. Further information on research design is available in the Nature Research Reporting Summary linked to this article.

## Data availability

The authors declare that all data supporting the findings of this study are available within the article and its Supplementary information files or from the corresponding author upon reasonable request. Source data are provided with this paper. The raw confocal micrographs of embryos expressing membrane marker, and the segmentation results and standardized morphological atlas generated in this study are available in the figshare repository: https://doi.org/10.6084/m9.figshare.12839315.

## Code availability

The deconvolution procedure on the raw confocal microscopies is implemented with commercial software Scientific Volume Imaging (SVI) Huygens Suite (https://svi.nl/HomePage). Codes for *C. elegans* embryo segmentation and subsequent systematic analsysis are available in the github repository (https://github.com/cao13jf/CShaper.git).

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

## Acknowledgements

The *C. elegans* strains were provided by the *C. elegans* Genetic Center (CGC), which is funded by the National Institutes of Health, Office of Research Infrastructure Programs, Grant P40 OD010440. This work was supported by the Hong Kong Research Grants Council (HKBU12100118, HKBU12100917, HKBU12123716, N_HKBU201/18, C1007-15G), the HKBU Interdisciplinary Research Cluster Fund, the Ministry of Science and Technology of China (2015CB910300), the National Natural Science Foundation of China (91430217), the Hong Kong Innovation and Technology Commission (ITC), and the Hong Kong Institute for Data Science.

## Author contributions

H.Y., Z.Z., and C.T. conceived and coordinated the study. J.C. designed the cell segmentation algorithm, G.G. analyzed the segmentation results, and V.W.S.H. designed the transgenic vector and made the transgenic lines expressing membrane labels. M.K.W. and L.Y.C. performed genetic crossings, imaging, and embryo curation. Z.Z. provided reagents and experimental methods. J.C. and G.G. wrote the paper, and H.Y., Z.Z., and C.T. revised the paper. All authors reviewed the results and approved the final version of the paper.

## Competing interests

The authors declare no competing interests.
