## [Peer Review File · Nature Communications]

Reviewers' Comments:

Reviewer #1:

Remarks to the Author:

"Establishment of morphological atlas of *Caenorhabditis 1 elegans* embryo with cellular resolution using deep-learning-2 based 4D segmentation" reports on a deep learning based method for single cell segmentation that was used to create an atlas of *C. elegans* development from the 1-cell to 350-cell stage. The authors created a *C. elegans* line that expressed both a nuclear and a membrane marker and created an annotated dataset for segmentation. Using this dataset, the authors then developed a deep learning method for single cell segmentation. This method is similar to the deep watershed method first reported by Urtasun et al in 2017, as it uses deep learning to predict a distance map directly from the images (although here the authors use a 3D deep learning model). The authors then use Delauney triangulation to seed a watershed process to create segmentation masks for the whole cell, not just the nucleus. The authors use this segmentation algorithm and pair it with existing cell tracking approaches to profile cellular morphologies as embryos grow from the single-cell stage to the 350-cell stage.

While there are several things to like about this paper (training data, quantitative single cell analysis, an open repository for all of the source code), from a deep learning perspective the novelty of the work presented is somewhat limited. It is not clear to me that this work is suited for a general audience. Deep learning is increasingly a commonplace approach to segmentation (see the recent review by Moen et al (<https://www.nature.com/articles/s41592-019-0403-1>) and using deep learning to predict a distance map has been done before (see Urtasun et al, <https://arxiv.org/ftp/arxiv/papers/1803/1803.10829.pdf>, <https://ieeexplore.ieee.org/abstract/document/8759242/authors#authors>) for nuclear and cytoplasmic data in both 2D and 3D. While there are some differences in the approach the authors have taken from these prior works (particularly when it comes to post processing), I think it would be better for the readers if this work were framed as an incremental advance from the algorithmic perspective. The data annotation the authors undertook is commendable, as is making the dataset public. However, the scale of the effort was not clear from the manuscript. Were 1000 cells annotated? 10,000? It is unclear from the manuscripts' current form the scale of this effort, which is unfortunate. Training data is sorely needed in this space, but it is hard to judge how much the authors have contributed to this problem. While the analysis of cellular morphology during development was interesting, I leave the judgement of its importance to other reviewers.

Other areas for improvement for the paper are given below.

- The benchmarking of the algorithm's performance could be substantially improved. In particular, benchmarking object based errors (number of merge/split errors etc) as was done previously (<https://www.biorxiv.org/content/early/2018/06/16/335216>) would be a considerable improvement.
- The author's GitHub repository is missing a Dockerfile. This would substantially increase the reproducibility of the author's work.
- The authors should make sure the training data contains relevant metadata like pixel dimensions.
- The size of the annotated dataset (number of cells) is not clear - this should be reported.
- I found the method description to be confusing and opaque. The author's should include a better description of how Delauney triangulation works and how it is used - a figure would definitely be informative.
- The paper contains numerous grammar and stylistic errors that need to be improved (e.g. Segmented region without any nucleus -> The segmented regions that did not contain nuclei, etc.). These errors are too numerous for me to list all of them, but a round of revisions with a copy editor would fix most of them and greatly improve the papers readability.

Reviewer #2:

Remarks to the Author:

This manuscript deals with the important problem of determining which cells contact each other (potentially enabling cell signaling or directed cell movements) during embryogenesis in the widely used model organism *C. elegans*. The quality of written English is quite poor. Nearly every sentence has at least one grammatical error. Most paragraphs are poorly structured, for example many begin with something like "In Figure X, Results Y are displayed" – instead you should start the paragraph with a topic sentence explaining why we should care about Results Y. The text is littered with unnecessary details, while critical conceptual points are not mentioned. This made it very challenging to thoroughly review the results, but I have tried my best. I understand that the authors are not native English speakers, but either they should find a native speaker to carefully edit their papers or hire an independent editor for this purpose before submission in the future.

The authors and others had previously inferred cell contacts from nuclear positions, but this paper adds to that by directly observing cell boundaries with a fluorescent membrane label. They develop a deep learning based method to identify cell boundaries from these fluorescent images (CShaper), and train this method on a gold standard set of manually validated/corrected boundaries. CShaper appears to perform better than other methods for boundary segmentation on both the gold standard dataset, and on the (appropriate) 'consistency over time' metric. If correct, this would be a useful advance that could lead to new insight in *C. elegans* research, and be applied in other systems. They further conclude (from just 4 embryos) that there is very little variability in cell volume and surface area between embryos for the same cell. Finally, they look in detail at shape changes and cell contacts in the daughters of the ABp cell. Other than the variability question, no new biology is studied or reported – if this would be done it would strengthen the impact of the paper.

Some questions/concerns:

Why were only 4 embryos used? It seems like the extra work to analyze a few more embryos would add substantially to confidence in the low-variability conclusion. Also, I don't think this conclusion is well demonstrated since the correlation in Fig. 3D/E (page 13 lines 1-5) is almost completely due to the small number of very large cells in the earliest embryos (which should also be the easiest to segment). It would be better to display some normalized measure of variability such as the Coefficient of Variation of each cell.

Are there cases where the surface area is higher than expected given the volume (suggesting an especially irregular shape)? If so (or if not) this would be worth reporting.

The application to plant data should be reported in the results, not discussion.

7:7 "voxel size around $0.22 \times 0.22 \times 0.22 \mu\text{m}$ " this voxel size is likely less than the optical resolution, especially in Z. Does this impact the accuracy of boundary segmentation? What was the original resolution before "resizing"?

7:18-19. Please explain acronyms in text briefly (3DMMS, ITK-SNAP)

7:20 "Nucleus image is imposed to prevent invalid gaps between cells" – it is not at all clear how this works or what it means (also this is grammatically incorrect)

7:26 "Companied" is not a word

10:14/17 – I'm not sure from the text the purpose of the NaN values, but regardless I think the way the matrix is described is too focused on details and should instead describe this matrix more conceptually

10:35 – it is unclear to me that the results support that we can "safely conclude" that Cshaper has superior ability given the relative lack of diversity of image and tissue types examined. Another possibility is that the authors are not as good at optimizing parameters, etc for the other methods as they are for their own method.

12:7 why "proposed"?

13:19 the authors should justify the use of 1/36 cell area in the text

Figure 4 – it is unclear to my eyes how to see the features described in the text – maybe these could be highlighted, or even better, illustrated with an accompanying cartoon.

15:21 remove the word “remarkable”

22:21 what is done with the segmented regions with no nuclei? Are they added to an adjacent cell, or considered to be extracellular space?

Response to the Referee # 1

General comments:

Comment 1:

Establishment of morphological atlas of Caenorhabditis elegans embryo with cellular resolution using deep-learning-based 4D segmentation” reports on a deep learning based method for single cell segmentation that was used to create an atlas of C. elegans development from the 1-cell to 350-cell stage. The authors created a C. elegans line that expressed both a nuclear and a membrane marker and created an annotated dataset for segmentation. Using this dataset, the authors then developed a deep learning method for single cell segmentation. This method is similar to the deep watershed method first reported by Urtasun et al in 2017, as it uses deep learning to predict a distance map directly from the images (although here the authors use a 3D deep learning model). The authors then use Delauney triangulation to seed a watershed process to create segmentation masks for the whole cell, not just the nucleus. The authors use this segmentation algorithm and pair it with existing cell tracking approaches to profile cellular morphologies as embryos grow from the single-cell stage to the 350-cell stage.

Response:

We thank the referee for the comments.

Comment 2:

While there are several things to like about this paper (training data, quantitative single cell analysis, an open repository for all of the source code), from a deep learning perspective the novelty of the work presented is somewhat limited. It is not clear to me that this work is suited for a general audience. Deep learning is increasingly a commonplace approach to segmentation (see the recent review by Moen et al (<https://www.nature.com/articles/s41592-019-0403-1>) and using deep learning to predict a distance map has been done before (see Urtasun et al, <https://arxiv.org/ftp/arxiv/papers/1803/1803.10829.pdf>, <https://ieeexplore.ieee.org/abstract/document/>

8759242/authors - authors) for nuclear and cytoplasmic data in both 2D and 3D. While there are some differences in the approach the authors have taken from these prior works (particularly when it comes to post processing), I think it would be better for the readers if this work were framed as an incremental advance from the algorithmic perspective.

Response:

In line with the methods referred to above (Urtasun et al.⁴⁴, Wang et al.⁴³, Eschweiler et al.⁴²), the distance map does get involved in the pipeline and serves as an indispensable part of CShaper. While profiling CShaper as an incremental advance, we would like to emphasize our contribution adopted for the segmentation of *C. elegans* embryo from the algorithmic perspective:

- Compared to these methods, the distance map in CShaper is designed and predicted in a different way. We threshold the distance to an upper maximum when the region locates far away from the membrane (Fig. R1). It is consistent with the fact that these regions are less discriminative for the prediction of continuous distance, thus they should be treated as the same. Hopefully, it helps the deep learning method to deal with cell size variation and thin membrane flexibly.
- The distance map is utilized to facilitate membrane segmentation rather than being a topographic map in the watershed algorithm. (1) Both Urtasun et al. and Wang et al. use the predicted distance map in the watershed segmentation directly (Fig. R2). Such strategy, however, suffers from the distributed regional errors during the flooding step. We

Figure R1 | Difference of sliced distance maps processed by Urtasun et al., Wang et al. and CShaper. (a) Raw membrane image. (b) Membrane annotation. (c) Distance map used by Wang et al. (d) Distance map used by Urtasun et al. (e) Distance map in CShaper. Translucent binary membrane is overlaid in (a), (b) and (c).

Figure R2 | Usage of distance map in Urtasun et al., Wang et al. and CShaper. Given a semantic segmentation, Urtasun et al. extract the boundary of each instance based on the distance map. Then each separated region is treated as one object, which can also be regarded as center-seeded watershed segmentation. Wang et al. utilized two separated networks to generate continuous distance map and seeds, respectively. Differently, after identifying the boundary membrane, CShaper derives seeds with a more reliable seeding procedure to prevent over- and under-segmentation.

Figure R3 | Comparison results on methods proposed by Urtasun et al. (DeepWatershed), Wang et al. (SingCellDetector), Eschweiler et al (3DUNet) and Ours (CShaper). (a) Dice ratio and (b) Hausdorff distance.

experimentally confirmed this by applying a post watershed transformation stage with the nuclei and predicted map being the seeds and the topology, respectively. It shows that CShaper can overcome the defects in the predicted map (Fig. R3). (2) Eschweiler et al. proposed to interpret the multi-instance segmentation as three-class semantic problem, including background, cell centroids and membrane, all of which can be regarded as the

Figure 1 | Benchmarking of segmentation results. (a, b, c) Evaluations based on manual annotations of cells in three wild-type samples (01-03) with seven time points per embryo. (a) The dice ratio of the segmentations generated by 3DUNet, CellProfiler, FusionNet, RACE, SingleCellDetector, B-CShaper and CShaper. Cell numbers were averaged at corresponding time points for each of the three embryos. (b) The average Hausdorff distance between the segmentation results produced by these methods and the ground truth for each sample. (c) Object-level F_1 scores for different IoU thresholds. (d, e) Statistics describing an additional 17 samples (05-21) imaged and segmented spanning the 4- to 350-cell stages. (d) Distribution of cell volume inconsistency coefficient (ρ_c) over time (t_c). (e) The number and ratio of lost nuclei over developmental time in the 17 embryos, where the last time point of the 4-cell stage is set as the starting time point. Each color represents an individual embryo. Solid and dashed lines denote the total number of cells that were successfully (total) and unsuccessfully (lost) segmented, respectively.

naïve binary segmentation. As discussed in the manuscript (Section *Methods – Distance constrained learning*), binary classification is prone to segmentation leakage when the membrane signal is too weak to be discriminated. This is a common case in *C. elegans* imaging because of the low fluorescent density and laser attenuation. The limitation of binary classification is also confirmed by comparison between binary CShaper (B-CShaper) and CShaper in Figs. 2a, b, c in the manuscript. (3) In CShaper, although the discrete

distance map is predicted, only a single class, the membrane (middle surface), is used in the following procedures. The reason why we predict the map is to impose shape features to the binary membrane mask so that DMapNet has the chance to learn morphological changes. The lost membrane and segmentation leakage can then be excluded.

- *Automatic seeding with grouped minimum remedies the defects of over- and under-segmentation that commonly exists in watershed transform.* Although watershed algorithm prevails in instance segmentation, inappropriate seeds can induce object-level errors. In CShaper, an effective graph-based clustering is proposed to reduce over- and under-segmentation (see Section *Methods - Watershed segmentation with automatic seeding*).
- *Pseudo-3D data flow.* 3D deep learning is desirable because of its superior ability in feature extraction. Theoretically, 2D network can be easily transformed to 3D version, followed with increased computational consumption and training data. However, given the limited spread depth of single fluorescent molecule and thickness of the membrane on the one hand, and the application with limited computation resource on the other hand, we achieve a compromise between 3D network and segmentation performance with a pseudo-3D convolution, where one 3D convolutional kernel is approximately realized through two consecutive 2D kernels. By comparing CShaper and 3DUnet (Figs. 2a, b, c), we note CShaper can provide competitive result as a 3D network.

Because the framework of deep watershed (Urtasun et al.) is designed for *Cityscapes instance level segmentation task* (<https://www.cityscapes-dataset.com/examples/>), a pretrained model PSPNet is required to extract the binary mask first. There exists substantial differences in either the dataset (2D city landscape vs 3D fluorescence image) or the segmentation framework (preliminary mask plus instance segmentation vs end-to-end instance segmentation), therefore deep watershed can be hardly applied on segmenting the embryo dataset. Here we only discussed the insights of deep watershed by transforming critical components in CShaper, including the loss function, distance map and seeding procedure. The comparison with this variant deep watershed is excluded in the manuscript. All the methods discussed here have been added into the reference list.

Apart from the algorithmic part, this work also serves general audience in biological fields, especially for *C. elegans* research. Sulston et al. profiled the stereotypic developmental programs of *C. elegans* manually and inspired biologists to use *C. elegans* for a wide range of topics in biological study. A lot of researchers paid a great effort in rebuilding the morphogenetic process by segmenting cell membranes (e.g., Azuma et al. 2017, Wang et al. 2019), but so far, a quantitative 4D morphological atlas with cellular resolution and identity has not been established yet. Thus, many recent works about morphogenesis still rely on nucleus-based data (e.g., Li et al.

2019, Jelier et al. 2016). Our work meets the needs of of high-quality morphological data with a general interest.

Comment 3:

The data annotation the authors undertook is commendable, as is making the dataset public. However, the scale of the effort was not clear from the manuscript. Were 1000 cells annotated? 10,000? It is unclear from the manuscripts' current form the scale of this effort, which is unfortunate. Training data is sorely needed in this space, but it is hard to judge how much the authors have contributed to this problem. While the analysis of cellular morphology during development was interesting, I leave the judgement of its importance to other reviewers.

Response:

To make our contribution on the training data clear, we have now added Supplementary Tables S1 *Experimental information of the wild-type embryos used in this study* and S10 *Wild-type embryo samples used in training and evaluation* to the Supplementary material. For the training data, it took almost two months to annotate 4572 3D cells in a semi-automatic way, which is now described in the Section *Methods - Manual annotation of cell*. Besides, based on the training data, we applied CShaper to segment 17 embryos at single-cell level as a ready-to-use morphological resource. Using StarryNite and AceTree, 656 cells with specific identity were reproducibly annotated for each embryo, including 322 cells with complete lifespan recorded (Supplementary Tables S4 *List of cells present in each embryo* and S5 *List of cells segmented in each embryo*, Section *Methods - Nucleus tracing and lineaging*). All the datasets are also available publicly on our CShaper website [<https://bit.ly/31rDpUk>] with complete resource information.

Specific comments:

Comment 1:

The benchmarking of the algorithm's performance could be substantially improved. In particular, benchmarking object based errors (number of merge/split errors etc) as was done previously (<https://www.biorxiv.org/content/early/2018/06/16/335216>) would be a considerable improvement.

Response:

Evaluation benchmarks used by Caicedo et al. include

- (1) Object-level F_1 score,
- (2) Fraction of missed nucleus at threshold IoU=0.7,
- (3) Split and merge errors,

- (4) Distribution of errors with respect to cell size,
- (5) Performance changing with the number of training images, and
- (6) The variety of data improves the performance.

Because the purpose of this work is two-fold, segmenting embryo and constructing morphological atlas, we concentrate on benchmarking CShaper from both algorithmic and biological perspectives. Therefore, we used following evaluations:

- (a) Dice (*pixel-level*): the ratio of the overlapped regions between the prediction and the ground truth (Fig. 2a),
- (b) Hausdorff distance (*pixel-level*): the largest distance of two paired pixels (Fig. 2b),
- (c) F_1 score (*object-level*): object-based segmentation F_1 score under different IoU scores (Fig. 2c),
- (d) Volume inconsistency (*object-level*): the conservation of cell size in consecutive frames (Fig. 2d),
- (e) Cell loss ratio (*object-level*): the ratio of cells that fail to be segmented (Fig. 2e),
- (f) Verification on previous experimental knowledges:
 - (i) (*object-level*) power relationship between cell cycle length and cell volume (Supplementary Fig. S5),
 - (ii) (*object-level*) known reproducible cell-cell contacts with signaling transduction (Supplementary Table. S7).

The complete *cell lineage* with cellular shape is involved in evaluations (d), (e) and (f). Although all the compared methods can segment individual frames, only CShaper is able to generate the *cell lineage*, including cell morphological trajectory, cell division timing and specific cell identity. Therefore, after comparing CShaper with other methods in (a), (b) and (c),

Figure S5 | Verification of the power relationship between cell cycle length and cell volume during development. Cell volume and cell cycle length of (a) AB and MS cells, with a power exponent ≈ -0.293 ; (b) C and P cells, with a smaller power exponent ≈ -0.363 . The insets denote the same data with a log-log scale coordinate system.

we evaluated (only) CShaper in evaluations (d), (e) and (f) to investigate its segmentation performance on the morphological atlas.

Comment 2:

The author's GitHub repository is missing a Dockerfile. This would substantially increase the reproducibility of the author's work.

Response:

The *Dockerfile* is now publicly available in the code repository.

Comment 3:

The authors should make sure the training data contains relevant metadata like pixel dimensions.

Response:

Now the metadata is embedded into the nifty file as the image header, which includes resolution, dimension, size, etc.

Comment 4:

The size of the annotated dataset (number of cells) is not clear - this should be reported.

Response:

To clarify the annotated data, Supplementary Tables S1 *Experimental information of the wild-type embryos used in this study* and S10 *Wild-type embryo samples used in training and evaluation* listed all the data involved in the manuscript. The details can also be found in the response to Referee #1 General comment 3.

Comment 5:

I found the method description to be confusing and opaque. The authors should include a better description of how Delauney triangulation works and how it is used - a figure would definitely be informative.

Response:

We have rewritten the Section *Methods*. For the automatic seeding step based on Delauney triangulation, the corresponding part is changed as follows,

“Watershed segmentation with automatic seeding. Given the distance map Ψ , watershed segmentation is well suited for separating individual cells. Although promising, direct application of watershed transformation to the map suffers from over-segmentation, where a single cell is split into multiple regions. Here, we proposed an automatic seeding procedure to facilitate the cellular segmentation. Associated with Delaunay triangulation, automatic seeding aims to detect appropriate seeds from the membrane mask for watershed algorithm.

Figure S9 | Automatic seeding based on the Delaunay triangulation. The red membrane contour corresponds to the K -th class predicted by DMapNet. Orange points denote the local maximum of the distance transformation, treating the contour as the background. (a) Local maxima s_1, \dots, s_{10} are connected as a graph with the blue edges indicating the Delaunay triangulation data. The color depth denotes the edge weights. (b) All edges with a weight higher than the OTSU threshold were removed. Grouped local maxima are connected to form one seed (orange lines).

The K -th class in Ψ was regarded as the membrane mask Φ^p . By selecting the background as the target voxel, Euclidean distance transformation (EDT) was applied to Φ^p , yielding \mathcal{M}^p . All local H -minima in \mathcal{M}^p were denoted as $\mathbf{S} = \{s_i\}_{i=1, \dots, L}$, where L was the number of local minima. A weighted graph \mathbf{G} was constructed to cluster s_i 's that belong to one cell or background. Edges $\mathbf{E} = \{\mathbf{E}_1, \mathbf{E}_2\}$ in \mathbf{G} came from two sources: one was the Delaunay triangulation on \mathbf{S} , noted as \mathbf{E}_1 , and the other was the edges, \mathbf{E}_2 , among all local minima located on the boundary of the volume. The weight of edge \mathbf{e}_{ij} was defined as

$$W(\mathbf{e}_{ij}) = \begin{cases} \sum_{(x,y,z) \in \mathbf{e}_{ij}} \mathcal{M}^p(x, y, z), & \mathbf{e}_{ij} \in \mathbf{E}_1 \\ 0, & \mathbf{e}_{ij} \in \mathbf{E}_2 \end{cases} \quad (7)$$

where $(x, y, z) \in \mathbf{e}_{ij}$ represents all points on the edge \mathbf{e}_{ij} . One edge was removed from \mathbf{E} if the corresponding weight was greater than the OTSU⁶⁷ threshold on W . Consequently, vertexes \mathbf{S} were clustered based on their connectivity. As opposed to inspecting each minimum, we treated each cluster, possibly including multiple minima, as one seed. Such group seeded watershed transformation on \mathcal{M}^p reduces under- or over-segmentation errors. A schematic description was also provided (Supplementary Fig. S9).”

Comment 5:

The paper contains numerous grammar and stylistic errors that need to be improved (e.g. Segmented region without any nucleus -> The segmented regions that did not contain nuclei, etc.). These errors are too numerous for me to list all of them, but a round of revisions with a copy editor would fix most of them and greatly improve the papers readability.

Response:

After careful revision and proofreading by a commercial service, the writing quality is now substantially improved.

Response to the Referee # 2

General comments:

Comment 1:

This manuscript deals with the important problem of determining which cells contact each other (potentially enabling cell signaling or directed cell movements) during embryogenesis in the widely used model organism C. elegans. The quality of written English is quite poor. Nearly every sentence has at least one grammatical error. Most paragraphs are poorly structured, for example many begin with something like “In Figure X, Results Y are displayed” – instead you should start the paragraph with a topic sentence explaining why we should care about Results Y. The text is littered with unnecessary details, while critical conceptual points are not mentioned. This made it very challenging to thoroughly review the results, but I have tried my best. I understand that the authors are not native English speakers, but either they should find a native speaker to carefully edit their papers or hire an independent editor for this purpose before submission in the future.

Response:

In order to present our work clearly, we have almost rewritten the manuscript and introduced diagrammatic figures and explanatory tables based on these constructive comments. Now the paragraphs are rearranged as follows,

- Abstract
- Introduction
 - Importance of cellular morphology from biological perspective
 - Difficulties in cellular morphology study
 - Traditional methods in cell segmentation
 - Deep learning based methods in cell segmentation

- Results
 - Overview on CShaper and the morphological atlas
 - Performance of CShaper
 - Comparison with other methods on three criteria, *Dice*, *Hausdorff distance* and *F1 score* based on ground truth annotations
 - Performance evaluation on *volume inconsistency and cell loss ratio* without ground truth annotations
 - Establishment of Morphological Atlas
 - Normalization and standardization on the embryo samples
 - Data quality and reproducibility in successfully segmented cells
 - *C. elegans* Biology Revealed by the Atlas
 - Cell size and its developmental accuracy
 - Cell-cell contact and physical requirements for signaling
 - Cell shape irregularity
 - Reproducibility and variability of cellular morphological dynamics
- Discussion
 - Verification of biology revealed in previous study
 - Cavities inside the *C. elegans* embryo
 - Applications to plant tissue image data
 - Constraints of CShaper
 - Brief summary
- Methods
 - *C. elegans* strain
 - Image acquisition
 - Nucleus tracing and lineaging
 - Manual annotation of cell
 - Distance constrained learning
 - Network structure DMapNet
 - Watershed segmentation with automatic seeding
 - Cell tracing and identification based on the segmentation
 - Standardization of embryo samples
 - Definition of effective cell-cell contact

Comment 2:

The authors and others had previously inferred cell contacts from nuclear positions, but this paper adds to that by directly observing cell boundaries with a fluorescent membrane label.

They develop a deep learning based method to identify cell boundaries from these fluorescent images (CShaper), and train this method on a gold standard set of manually validated/corrected boundaries. CShaper appears to perform better than other methods for boundary segmentation on both the gold standard dataset, and on the (appropriate) ‘consistency over time’ metric. If correct, this would be a useful advance that could lead to new insight in *C. elegans* research, and be applied in other systems. They further conclude (from just 4 embryos) that there is very little variability in cell volume and surface area between embryos for the same cell. Finally, they look in detail at shape changes and cell contacts in the daughters of the ABp cell. Other than the variability question, no new biology is studied or reported – if this would done it would strengthen the impact of the paper.

Response:

We further explored four biological problems quantitatively. Cell shape irregularity and dynamics are added according to Referee #2 Specific comments 2.

1. Cell size and its developmental accuracy

We had evaluated the reproducibility and variability of cell size among individual embryos, including cell volume and cell surface area. We thank the referee for the suggestion of adding more samples to support our statistical finding. We finally collected a total of 17 wild-type embryo samples (only 4 samples used previously) to draw the conclusions:

- (1) The cell size reproducibility among individual embryos is highly controlled at single-cell level (goodness of fit > 0.99 , CV < 0.2).
- (2) The *C. elegans* embryo can tolerate a natural variability in cell size of each cell, as large as 0.2 in coefficient variation.

The results are illustrated in Fig. R4 (Figs. 3d, e in the manuscript).

Figure R4 | Reproducibility and variability of (a) cell volume and (b) cell surface area. The main graphs are plotted using proportionally normalized data from all 322 cells with a complete lifespan (Supplementary Table S4). Top-left insets: data graphed using original data prior to normalization. Bottom-right insets: variation coefficients of each cell among the 17 embryos in the normalized data.

2. Cell-cell contact and physical requirements for signaling

Based on previous studies, we reexamined the known cell-cell signalings and explored the limit of physical requirements on contact area as well as contact duration. We set up three specific filter criteria to define an effective cell-cell contact (Fig. 3f, Supplementary Table S6). In some embryo samples, two contacts with known signaling transduction did not pass through completely because they failed with either the threshold of contact area or contact duration (Supplementary Table S7). It reveals that intercellular signaling relies less on these two conditions than the previous assumptions². The observed looser physical requirements (i.e., relative contact area $< 1/48 \approx 2.08\%$, contact duration ≤ 1 time point ≈ 1.5 min) indicate that much more cell-cell contacts may have potential to be functional.

- (1) Even though we set up a threshold for relative cell-cell contact area (2.08%) that is lower than the value previously used (6.5%)², the contact areas in both C-ABar (Wnt signaling) and MSapp-ABplpapp (Notch signaling) were found to be even smaller and did not pass through the criteria in several embryo samples.
- (2) The contact duration of MSapp-ABplpapp was found to last only one time point (≈ 1.5 min) in an embryo sample.

3. Cell shape irregularity

We quantified the shape irregularity of 322 cells by a dimensionless surface-to-volume ratio η with following conclusions.

- (1) The cells always resemble octahedron, cube and tetrahedron, but not as round as sphere, icosahedron or dodecahedron (Figs. 4a, b).
- (2) Cell shape irregularity is associated with cell lineage and cell fate. In most cases, the AB progenies are rounder than the ones in P1 (Fig. 4b). An outstanding example is the

Figure S4 | Bilaterally symmetric intestine cells at E8 stage. The eight intestine cells present at this stage are illustrated from the view of image shooting perspective from embryo Samples 11 (left) and 12 (right). Each color represents an individual E cell as indicated.

bilaterally symmetric gut-like structure formed by intestine precursor cells E8, which have numerous spikes or wrinkles on their surfaces (Supplementary Fig. S4).

- (3) MS and ABpl are the two cells with the most severe deformation among the 322 cells inspected, while the other cells at the same time are much regularly shaped (Fig. 4b, c).

Figure 4 | Cell shape irregularity during *C. elegans* embryogenesis. (a) Dimensionless irregularity scores, η , for a sphere, icosahedron, dodecahedron, octahedron, cube and tetrahedron. (b) Distribution of cell shape irregularity across development as a cell lineage tree. The color bar denotes the level of shape irregularity, with the two most irregular cells, ABpl and MS, indicated by red triangles. (c) Time-lapse 3D cell shape dynamics of ABpl (blue), MS (green), ABal (pink) and P3 (red) derived from embryo Sample 05. Horizontal and vertical axes represent anterior-posterior (A-P) and dorsal-ventral (D-V) axes, respectively. Dynamics of 3D shape is shown over developmental timings (mean \pm standard deviation) indicated at the top (Supplementary Table S3).

4. Reproducibility and variability of cellular morphological dynamics

Several studies characterized the accuracy of cellular behaviors during *C. elegans* embryogenesis, especially for the cell migration^{1,8,9,51}. Here we used ABp, ABpl and ABpr cells to address the loss of accuracy in some morphological behaviors, which help us identify the essential activity

Figure 5 | Morphological dynamics at single-cell resolution. Time-lapse 3D cell shapes during *C. elegans* embryogenesis for ABp (purple) and its daughter cells, ABpl (blue) and ABpr (cyan). Horizontal and vertical axes represent anterior-posterior (A-P) and dorsal-ventral (D-V) axes, respectively. **(a)** Morphological dynamics are shown as in Fig. 4c for the time points immediately before and after AB2 divisions (columns 1-2), before and after EMS division (columns 3-4), before and after P2 division (columns 5-6) and before AB4 divisions (column 7) (Supplementary Table S4). The first row shows the cell position (nuclei) distributions obtained from all 46 wild-type embryos (05-50). The second to seventh rows show reconstructed cell morphologies from the 17 embryo samples segmented with membrane markers. Embryo Samples 09, 11, 13, 15, 16 and 20 were selected to demonstrate the morphological dynamics of ABp and its daughter cells. Note that in the second and third columns the division orientation of ABp is marked by solid and dashed lines for unseparated and separated daughter cells, respectively. In the sixth column, sharp humps on the ABpl's surface are highlighted by black arrows. In the last column, ABpl and ABpr can be observed in a connecting state, intermediate state or separating state, exemplified by Samples 15 and 16, Samples 11 and 20, Samples 09 and 13, respectively. **(b)** Position and morphology of ABpl (blue) and ABpr (cyan) at the time point prior to AB4 division. The embryo samples with ABpl-ABpr cell pairs in a connecting state are highlighted by black squares.

for development. We observed the time-lapse morphologies of ABp, ABpl and ABpr cells, and investigated which features are reproducibly observed among individuals and which are not.

- (1) The division orientation of ABp has a bias toward the dorsal-ventral direction in some embryo samples. However, this feature is not completely reproducible, indicating that the anteroventral movement of ABpl does not rely on the division of its mother (Fig. 5a).
- (2) The final contact states of ABpl and ABpr differ from embryo to embryo. They can be connected, separated or in an intermediate state, suggesting that no essential interaction is undergoing between these two cells (Figs. 5a, b).

Specific comments:

Comment 1:

Why were only 4 embryos used? It seems like the extra work to analyze a few more embryos would add substantially to confidence in the low-variability conclusion. Also, I don't think this conclusion is well demonstrated since the correlation in Fig. 3D/E (page 13 lines 1-5) is almost completely due to the small number of very large cells in the earliest embryos (which should also be the easiest to segment). It would be better to display some normalized measure of variability such as the Coefficient of Variation of each cell.

Response:

To support our conclusions statistically, we finally collected a total of 17 embryos (4 embryos used previously) with nucleus lineaging and cell segmentation. Preliminary results are filtered through rigorous requirements before composing into the standard morphological atlas (Supplementary Table S4).

Here we describe Fig. R4 (Figs. 3d, e in the manuscript) and its data sources clearly. We take the cell volume as an example to explain the derivation of variation coefficients (i.e., coefficient of variation) (Fig. R4, bottom-right insets), and these procedures were applied to cell surface as well. For each embryo, we extracted the volumes of 322 specific cells which have complete lifespan recorded in the cell lineage (Supplementary Table S4). Thus, each cell gets 17 volume values respectively from the 17 embryo samples. For each cell, we calculated its average volume, and then obtained all the volume averages of the 322 cells. Finally, as shown in the main diagram (after normalization) and the top-left inset (before normalization) of Fig. R4, by using the volume averages of the 322 cells as a reference, we aligned each embryo onto it with proportional fitting. Because each cell's volume in the 17 embryos always approaches to the average value (i.e., $V_A \approx$

V_5) no matter whether normalization is applied, the low variation coefficient of each cell (< 0.20) and high correlation between individual embryos (> 0.99) is reliable.

Regarding the referee's suggestion on "normalized measure of variability such as the Coefficient of Variation of each cell", we provided the measure in the bottom-right insets of Fig. R4. The histogram represents the distribution of 322 variation coefficients from 322 specific cells respectively. Regarding the referee's comment on "...is almost completely due to the small number of very large cells in the earliest embryos", we emphasize that no matter a cell is small or large, its volumes in 17 embryo samples are always close to each other ($CV < 0.20$). At last, two conclusions were made as below,

- (1) The cell size reproducibility among individual embryos is highly controlled at the single-cell level (goodness of fit > 0.99 , $CV < 0.2$).
- (2) The *C. elegans* embryo accepts a natural variability in cell size for each cell, limited within 0.2 in coefficient variation.

Comment 2:

Are there cases where the surface area is higher than expected given the volume (suggesting an especially irregular shape)? If so (or if not) this would be worth reporting.

Response:

We thank the referee for the suggestion of analysis on cell shape irregularity. We have now introduced a dimensionless surface-to-volume ratio η to quantify the shape irregularity of 322 cells. A group of Platonic Polyhedras as well as the sphere are chosen to clarify how this coefficient reveals the shape irregularity monotonously. We constructed a lineage tree coupled with the η value for visualization, and reported the observations on shape irregularity at both the cellular and lineal levels (Fig. 4).

The corresponding content has been introduced in Referee #2 General comment 2.3, and added in Section *Results - C. elegans Biology Revealed by the Atlas - Cell shapes*.

Comment 3:

The application to plant data should be reported in the results, not discussion.

Response:

Because in Section *Results*, we concentrate on analyzing the segmentation results of *C. elegans* embryo from both algorithmic and biological perspectives, it is better to arrange *the application of CShaper on plant tissue data* into Section *Discussion*. Furthermore, when targeting on processing *C. elegans* embryos, we only briefly discuss the potential usage of CShaper on dataset other than the *C. elegans*.

Comment 4:

7:7 “voxel size around $0.22 \times 0.22 \times 0.22 \mu\text{m}$ ” this voxel size is likely less than the optical resolution, especially in Z. Does this impact the accuracy of boundary segmentation? What was the original resolution before “resizing”?

Response:

As shown in Fig. R5, the original intra-slice and inter-slice resolutions are $0.09 \mu\text{m}$ and $0.42 \mu\text{m}$ ($0.43 \mu\text{m}$ for training data), respectively. However, anisotropic resolution makes the stacked volumetric embryo seem be flattened artificially (see from lateral view in Fig. R5a). In order to promote the DMapNet (deep learning model used in CShaper) to learn the realistic shape of cells instead of the flattened one, we linearly resize the volume into an isotropic resolution $0.25 \mu\text{m}$. Such resolution reaches a compromise between two different directions and also reduce computational burden. Therefore, the resize procedure can improve the accuracy of boundary segmentation theoretically.

Figure R5 | Resize operation on raw image. Raw volumetric image (a) is resized into image (b) with isotropic resolutions in both intra-slice and inter-slice directions. Resolutions are annotated at corresponding directions before and after the resize process.

Comment 5:

7:18-19. Please explain acronyms in text briefly (3DMMS, ITK-SNAP)

Response:

We have now provided the full names (“3D membrane morphological segmentation” and “an interactive tool for semi-automatic segmentation of multi-modality biomedical images”) of the two tools in addition to references to the relevant publications.

Comment 6:

7:20 “Nucleus image is imposed to prevent invalid gaps between cells” – it is not at all clear how this works or what it means (also this is grammatically incorrect)

Response:

As noted in REF 57, a blastocoel inside the embryo, which consists of several cavities, forms gradually since 4-cell stage to prepare the inner space for the upcoming gastrulation. Here, the “invalid gaps” mean such kind of cavities. While automatic seeding in CShaper can detect nucleus-like seeds for each separated regions, the nucleus channel is used to check whether each partitioned region actually includes nucleus. If not, the empty region will be regarded as a cavity and excluded from the cell sets. A cavity example is shown in Fig. R6. The region inside the dotted circle is recognized as a cavity because no nucleus is found from nucleus channel inside this region.

Figure R6 | Segmentation cavity inside the embryo. Raw image (a) and overlaid segmentation (b) are shown in three side views. A cavity in segmentation (b) is annotated with a dotted circle.

Comment 7:

7:26 “Companied” is not a word

Response:

We have corrected this typo.

Comment 8:

10:14/17 – I’m not sure from the text the purpose of the NaN values, but regardless I think the way the matrix is described is too focused on details and should instead describe this matrix more conceptually

Response:

The description about the temporal consistency evaluation has been rewritten (Page 11: 5 ~ 14).

Comment 9:

10:35 – it is unclear to me that the results support that we can “safely conclude” that CShaper has superior ability given the relative lack of diversity of image and tissue types examined. Another possibility is that the authors are not as good at optimizing parameters, etc for the other methods as they are for their own method.

Response:

- Diversity of image and tissue types: (1) As indicated in the title, we mainly concentrate on processing *C. elegans* embryos and establish a morphological atlas/dataset for the relative researches. Therefore, most parts of the manuscript focus on the *C. elegans* samples whose invariant lineage empowers systematic developmental biology. Experimental result on plant tissue serves to explain the *potential* application of CShaper on similar dataset, which, at present, is not our focus. (2) Practically speaking, low image quality may impact the segmentation performance of CShaper, especially for the imaging *in vivo* over a long duration. For example, we tried to increase the time resolution (by 10 seconds), but CShaper failed to recognize boundary cells where the membrane is critically lost (Fig. R7). The worm strain and imaging technique substantially contribute to the promising performance and reproducibility of CShaper (see *Methods - C. elegans strain & Image acquisition*).

Figure R7 | Segmentation example of CShaper when the image quality is extremely low. (a) Raw image; (b) Cell segmentations. Lost cells are indicated with dotted circle.

- **Parameter optimization:** In order to explain all parameter optimization strategies, we have now added Supplementary Note 2 *Parameter settings in compared methods* and Table S2 *Implementation information of compared methods* as supplementary materials. Mostly, we followed the settings originally proposed in those compared methods to achieve optimal results. However, some components, such as data loader format, were revised to accommodate our dataset if necessary. We also provide project files and source code to reproduce our evaluation results.

Comment 10:

12:7 why “proposed”?

Response:

The typo is corrected now.

Comment 11:

13:19 the authors should justify the use of 1/36 cell area in the text

Response:

Now the threshold of relative contact area is corrected to 1/48 (1/36 in our previous draft), while the derivation steps remain the same. We attempt to evaluate how many neighbors a cell can have when their sizes are different to some extent. It is well known that each sphere is surrounded by

Figure S13 | Estimation of the sufficient threshold for effective cell-cell contact. (a) Radius ratio between neighboring cells over developmental time. Each color represents one of the 17 embryo samples (05-21). (b-d) Hexagonal closely-packed structures from a (b) 3D view, (c) side view and (d) front view. (e) A unit cell is replaced by n_R cells with radius r_{max} . The red sphere represents the center cell O_0 , and the blue spheres represent the 11 constant neighbors. The green spheres represent the new, smaller cells that replace the original cell, with a maximum radius, r_{max} , obtained from the 109 independent trials in the simulation. The r_{max} approaches $1/3$ when $n_R = 4$.

12 neighbors in a close-packed structure of equal-sized spheres, which in theory has the highest space occupancy and system stability⁶⁸. In the *C. elegans* embryo, the radius ratio between neighboring cells can reach up to 3:1 (Supplementary Fig. S13a). Thus, based on the hexagonal close-packed structure, we estimated the cell-cell contact area threshold by simulating how many cells with a radius ratio of up to $1/3$ can be accommodated within space formed by a unit cell with a radius of 1 (Supplementary Figs. S13b, c, d). As a uniform neighboring cell can be replaced by, at most, four smaller cells with a radius ratio of $1/3$ (Supplementary Fig. S13e), the relative contact threshold was set as $1/12 \times 1/4 = 1/48 \approx 2.08\%$ (Section *Methods - Definition of effective cell-cell contact*, Supplementary Note 3 *Computational estimation of contact area threshold* and Fig. S13).

Comment 12:

Figure 4 – it is unclear to my eyes how to see the features described in the text – maybe these could be highlighted, or even better, illustrated with an accompanying cartoon.

Response:

As shown in Fig. 5 (Fig. 4 in our previous draft), we have added labels to highlight the morphological features.

- (1) To show the orientational bias of ABp division, we added a black line on top of the cell structure(s) to align the division orientation. We also used a solid one for an unseparated ABp cell, and a dashed one for separated ABpl and ABpr cells. These labels can show the obvious bias that exists in Samples 11 and 20, but not in Samples 15, 16, 09, 13 (2nd and 3rd columns in Fig. 5a).
- (2) To show the severe deformation of ABpl, we added black arrows pointing at the sharp humps on its surface. These humps are identified qualitatively and can be easily recognized.

To show the physical contact state between ABpl and ABpr, we added red triangles pointing at the interface, the contact point and the gap between them for the connecting state, intermediate state and separating state, respectively. Besides, we added a sub-figure to show the ABpl-ABpr terminal structures in all the 17 embryos, with black squares highlighting the ones with considerable contact interface.

Comment 14:

22:21 what is done with the segmented regions with no nuclei? Are they added to an adjacent cell, or considered to be extracellular space?

Response:

The segmented regions with no nuclei are considered to be extracellular spaces, namely cavities inside the embryo. These cavities can be filtered out semi-automatically and considered as a type of embryo structure with specific biological meaning. The cavities were generated by the cooperation of cell polarization, cell adhesion and other processes, which is crucial for the gastrulation initiated at the 26-cell stage⁵⁷. Based on our segmentation results, a blastocoel

Figure S6 | Identification of the blastocoel inside *C. elegans* embryo at the 26-cell-stage. Three cavities are identified in embryo Sample 15 at time point 35. (a) Cross-sectional microscopy images (left) with segmentations (right). The view directions are as in Fig. S2. (b) Rendered 3D structure. Each color represents one specific blast cell identity. The inner cavities are colored black. D, dorsal; V, ventral; L, left; R, right; A, anterior; P, posterior.

consisting of three discrete cavities is shown in Supplementary Fig. S6. As these cavities can be captured by CShaper, at least partially, our segmentation pipeline enables quantitative study of such intercellular spaces and their underlying mechanisms and functions (Section *Discussion - Cavities inside the C. elegans embryo*).

Reviewers' Comments:

Reviewer #1:

Remarks to the Author:

The authors have satisfactorily addressed my concerns. It would be great if figures R1 and R2 made it into the supplement. While the benchmarking the authors have done is improved, it would be great if recall and precision were reported, in addition to the number of merge and splitting errors.

Reviewer #2:

Remarks to the Author:

I first want to say that this paper has improved more after review than almost any paper I have ever reviewed, I give the authors a lot of credit for really taking the reviews to heart and working to improve the manuscript. The writing is now very easy to follow. I think the explanations in the results of the technical aspects of the work are also now very clear, and the increased data quality (17 embryos instead of 4 for example) and more in depth analysis of variability in cell shape between embryos and cells adds a lot to the paper. The lineage specific and cell cycle specific variation in cell roundness is really an interesting finding.

I have a few additional minor suggestions but overall I am enthusiastic about this paper.

The observation that some known signaling interactions are very short/small surface area is cool, however the authors should mention the alternative (also interesting possibility) that other signaling cells could compensate in embryos with short/small contacts (for example, at the 8-cell stage several cells in addition to C are competent to induce a Wnt signal). It is also possible that these signaling events can occur over short distances mediated by diffusible ligands. Either way, it is unclear from the data whether the examples on page 16 represent false negatives or interesting new biology, so the authors should mention both possibilities.

I'd consider this optional but potentially quite interesting and would make a lot of sense to include in this paper: given the modestly low variation of cell volume (CV of 0.2), is their higher consistency of asymmetry between sisters (e.g. examples of consistent size-asymmetric division)?

5:18 - I would add image quality as something that needs to be balanced against phototoxicity etc here

12:3 - using the expectation of temporal consistency is a useful check on errors, as the authors state, but some types of errors might not be captured here because the image characteristics of a given nucleus are often fairly similar across time....

13:21 "demanded that 656 cells be consistently present" should be "identified 656 cells consistently present"?

18:15 - this section is very nice in general, but I wouldn't state that the S/V ratio indicates the cells are "mostly octahedron, cube and tetrahedron in shape" as a similar s/v ratio could be obtained for example by a sphere with one or more irregular projections or other more complex shapes.

Figure 4B - could the authors increase the line thickness for the earlier branches? It is very hard to see the color scale to compare between lineages otherwise. It would be helpful to have some complementary graph (for example scatter plot of S/V ratio for time color coded by lineage) to make the point about lineage differences more clearly.

19:25 - since "lamellipodia", "protrusion" and "filopodia" have distinct meanings, I would say these bumps are "consistent with lamellipodia, protrusions or filopodia"

Page 23 - I would also emphasize that since the reason for the decrease in cell segmentation accuracy at later stages is the small size of cells and membrane features relative to the imaging resolution, improved image resolution and quality could allow better performance of CShaper at these stages.

Response to the Referee # 1

Comment 1:

The authors have satisfactorily addressed my concerns. It would be great if figures R1 and R2 made it into the supplement. While the benchmarking the authors have done is improved, it would be great if recall and precision were reported, in addition to the number of merge and splitting errors.

Response:

We sincerely thank the referee for the comments and the constructive suggestions.

1. The *Figures R1* and *R2* have been moved to the Supplementary Information as *Supplementary Figures 13* and *14*.
2. Based on the suggestion, now we report more object-level scores in *Supplementary Table 2. Comparison of Object-Level Performance of Methods*. The *split*, *merge*, *recall* and *precision* are defined as

$$\begin{aligned} \textit{split} &= \frac{\# \textit{split cells}}{\# \textit{total cells}} \\ \textit{merge} &= \frac{\# \textit{merged cells}}{\# \textit{total cells}} \\ \textit{recall} &= \frac{\# \textit{true cells}}{\# \textit{true cells} + \# \textit{lost cells}} \\ \textit{precision} &= \frac{\# \textit{true cells}}{\# \textit{true cells} + \# \textit{false cells}} \end{aligned}$$

For each method, the score is averaged over all samples (3 embryos \times 7 time points = 21 samples). As suggested by Caicedo et al. (DOI: 10.1038/s41592-020-0733-z), the IoU 0.7 is used to determine whether a segmentation is a split/merged/true/false cell. Because the nucleus location is directly used as seeds during the segmentation stage, some methods, such as 3DUNet and CellProfiler, are less prone to the over-segmentation (split errors). However, their segmentations are heavily dependent on the availability of nucleus information.

Supplementary Table 2. Comparison of Object-Level Performance of Methods

Method	Split	Merge	Precision	Recall
3DUNet ⁵	0.21%	2.44%	86.96%	72.79%
CellProfiler ¹	0.24%	2.02%	93.01%	78.03%
FusionNet ³	4.32%	1.79%	72.91%	66.66%
RACE ²	2.95%	6.21%	74.44%	72.39%
SingleCellDetector ⁴	2.24%	3.19%	81.95%	80.11%
B-CShaper	7.13%	0.00%	84.67%	55.91%
CShaper	0.67%	0.31%	98.35%	97.28%

Note: All scores are calculated at 0.7 IoU between the ground truth annotation and automatic segmentation. A satisfactory method is expected to not only have relatively small split and merge errors, but also keep high precision and recall scores.

Response to the Referee # 2

Comment 1:

I first want to say that this paper has improved more after review than almost any paper I have ever reviewed, I give the authors a lot of credit for really taking the reviews to heart and working to improve the manuscript. The writing is now very easy to follow. I think the explanations in the results of the technical aspects of the work are also now very clear, and the increased data quality (17 embryos instead of 4 for example) and more in depth analysis of variability in cell shape between embryos and cells adds a lot to the paper. The lineage specific and cell cycle specific variation in cell roundness is really an interesting finding.

Response:

We sincerely thank the referee for the comments and the constructive suggestions.

Comment 2:

I have a few additional minor suggestions but overall I am enthusiastic about this paper.

The observation that some known signaling interactions are very short/small surface area is cool, however the authors should mention the alternative (also interesting possibility) that other signaling cells could compensate in embryos with short/small contacts (for example, at the 8-cell stage several cells in addition to C are competent to induce a Wnt signal). It is also possible that these signaling events can occur over short distances mediated by diffusible ligands. Either way, it is unclear from the data whether the examples on page 16 represent false negatives or interesting new biology, so the authors should mention both possibilities.

Response:

The corresponding sentence is changed into “*This could also be due to some other redundant signaling events which compensate this interaction. Alternatively, this physical interaction may be mediated by diffusible ligands in proximity as proposed previously⁵⁵. The false negative may be unavoidable using these specific requirements, because the actual sensitivity of intercellular signaling including both contact area and contact duration is still poorly understood.*”

Comment 3:

I'd consider this optional but potentially quite interesting and would make a lot of sense to include in this paper: given the modestly low variation of cell volume (CV of 0.2), is their higher consistency of asymmetry between sisters (e.g. examples of consistent size-asymmetric division)?

Response:

As suggested, we added the systematic analysis on size ratio between sister cells in parallel with the absolute cell size (Supplementary Figure 7, Supplementary Table 5). First, a total of 161 pairs of sister cells were derived from the 322 cells which have a complete lifespan. Second, we calculated their size ratios (i.e., volume ratios and surface area ratios) and variation coefficients among the 17 embryo samples. By statistics, two more conclusions were made as below:

- (1) The size ratio between sister cells is less variable than the absolute cell size. This statement is consistent for both cell volume and cell surface area, under measurements on average, minimum, maximum and maximum of 99% data (Supplementary Table 5).
- (2) There is no significant or strong correlation between the size ratios and their variations among 17 embryos. The Pearson correlation coefficient between cell size ratios and their variation coefficients among 17 embryo samples is smaller than 0.15, with respect to both cell volume and cell surface area. The weak correlation can not be claimed with enough confidence.

Supplementary Figure 7. Reproducibility and variability of size ratio between sister cells. A total of 161 pairs of sister cells among the 17 wild-type embryos were evaluated. **(a, d)** Distribution of ratios for volume **(a)** and surface area **(d)** compared to their averages ($V_{A,1} / V_{A,2} \geq 1$, $S_{A,1} / S_{A,2} \geq 1$). Each color represents an individual embryo. **(b, e)** Variation coefficients of each cell pair among the 17 embryos. The average and standard deviation of the variation coefficients are shown on the top right. **(c, f)** Correlation between cell size ratios and their variation coefficients among the 17 embryos. The Pearson correlation coefficient r ($r = \frac{\sum x_i y_i - \frac{1}{N} \sum x_i \sum y_i}{\sqrt{[\sum x_i^2 - \frac{1}{N} (\sum x_i)^2][\sum y_i^2 - \frac{1}{N} (\sum y_i)^2]}}$; x_i and y_i , the two groups of variables subjected to test; N , sample size) is shown on the top right, revealing no significant or strong correlation. Source data are provided as a Source Data file.

Supplementary Table 5. Variation Coefficient of Different Cell Size Properties.

Cell Size Property	Average \pm Standard Deviation	Maximum of 99% Data	Maximum	Minimum
Volume	0.0921 \pm 0.0336	0.1999	0.2836	0.0300
Volume Ratio Between Sister Cells	0.0850 \pm 0.0335	0.1670	0.1988	0.0265
Surface Area	0.0644 \pm 0.0232	0.1434	0.1883	0.0202
Surface Area Ratio Between Sister Cells	0.0596 \pm 0.0218	0.1165	0.1214	0.0195

The corresponding paragraph is changed into “For each of the 322 cells, the variation coefficients of cell size among individual embryos range between around 0 to 0.2, indicating that a considerable level of variability is tolerated for both cell volume and cell surface area (Figures 3d, e, Supplementary Table 5). Despite the relatively high variation coefficients, the size ratio between sister cells (161 pairs in total) is less variable than the overall cell size, suggesting a more precise control over the size ratio between sister daughter cells during cytokinesis

than over the absolute size of each daughter. This is the case for both cell volume and cell surface area, under all the four measurements tested (i.e., average, minimum, maximum and maximum of 99% data). For example, 99% of the individual cells have a variation coefficient of ≤ 0.1999 in cell volume, while 99% of the sister cell pairs have a variation coefficient of ≤ 0.1670 in volume ratio (Figures 3d, e, Supplementary Figures 7a, b, d, e, Supplementary Table 5). Moreover, there is no significant or strong correlation between the ratio of cell volume or cell surface area and their variations for the 322 cells (161 cell pairs) among 17 embryos (Supplementary Figures 7c, f).”

Comment 4:

5:18 - I would add image quality as something that needs to be balanced against phototoxicity etc here.

Response:

The corresponding sentence is changed into “However, a careful tradeoff between image quality and phototoxicity has to be made during 4D imaging.”

Comment 5:

12:3 - using the expectation of temporal consistency is a useful check on errors, as the authors state, but some types of errors might not be captured here because the image characteristics of a given nucleus are often fairly similar across time...

Response:

The expectation of temporal consistency (referred as volume inconsistency) is mainly designed to capture the volume differences of the same cell across time. It is expected that if a cell is wrongly segmented, the error should be randomly distributed, especially at the cellular boundary, leading to high volume variance. Consequently, the volume of the corresponding cell should have low temporal consistency.

However, some errors may still get missed with regard to the volume inconsistency. For example, an extreme situation is that a nucleus is lost in the segmentation during its whole lifespan, the volume inconsistency would also be low (because all volumes are 0), in which case the volume-based criterion fails to evaluate the segmentation performance. Fortunately, after calculating the ratio of cells that are completely lost within their lifespans in the segmentation results, we found that such extreme case only occupies a small fraction (0.44%) in 17 embryo samples. Over 80% of cells are successfully segmented with less than 10% of their lifespans missed. Therefore the volume inconsistency may serve as a reasonable criterion when annotating extensive 3D microscopies is difficult. We have added this note to the manuscript - “The ρ_c may fail to capture errors when a cell is lost in too many time points within its lifespan, but such case only occupies a small fraction in the 17 embryo samples (Supplementary Table 3). Over 80% of cells are successfully segmented with less than 10% of their lifespans missed.”

Supplementary Table 3. Cell Loss Ratio under Different Fractions of Lifespan.

Fraction of Lifespan	0.1	0.2	0.3	0.4	0.5	0.6	0.7	0.8	0.9	1.0
Cell Loss Ratio	19.96%	12.19%	7.96%	5.77%	4.48%	3.11%	2.32%	1.64%	0.95%	0.44%

Note: Fraction of Lifespan = 0.1 means that a cell is lost within at least 0.1 (included) of the whole lifespan. For example, if the length of one cell's lifespan is 50 time points, it would be lost in at least 5 time points.

Comment 6:

18:15 - this section is very nice in general, but I wouldn't state that the S/V ratio indicates the cells are "mostly octahedron, cube and tetrahedron in shape" as a similar s/v ratio could be obtained for example by a sphere with one or more irregular projections or other more complex shapes.

Response:

The corresponding sentence is changed into "The irregularity η ranges from 2.3479 to 2.8060 in the cells examined across their lifespans, while their temporal averages range from 2.4011 to 2.5934, suggesting that the cells are mostly deformed in a severe level similar to those in octahedron, cube and tetrahedron, and are not as round as a perfect sphere, icosahedron or dodecahedron (Figures 4a, b, Supplementary Data 1)."

Comment 7:

Figure 4B - could the authors increase the line thickness for the earlier branches? It is very hard to see the color scale to compare between lineages otherwise. It would be helpful to have some complementary graph (for example scatter plot of S/V ratio for time color coded by lineage) to make the point about lineage differences more clearly.

Response:

The line thickness has been increased to ~ 1.7 times of the previous value. As suggested by the editor, we further replaced the rainbow color bar by the one changing from green to magenta, which has higher contrast ratio and more distinguishable in the white background (Figure 4b).

To illustrate the difference of cell shape irregularity between AB and P1 lineages, we used a box plot to show the η values of the last generation of AB and P1 cells, which have a complete lifespan recorded (i.e., AB128, MS16, E8 and C8 cells). Then, we performed one-sided Wilcoxon rank-sum test for statistical comparisons. The η values of MS, E and C sublineages are all significantly larger than those of AB lineage. The remaining D sublineage, which is also derived from P1, is excluded because there are only 4 D cells in its last generation.

Figure 4b. Distribution of cell shape irregularity over development shown in a cell lineage tree, with the two most irregular cells, ABpl and MS, indicated by red triangle. The color scheme denoting the level of shape irregularity is shown on the right. Source data are provided as a Source Data file.

Supplementary Figure 8. Comparison of cell shape irregularity between AB (blue) and P1 (red) lineages, using the last generation of the 322 cells with a complete lifespan. All the sublineages of P1 (i.e., MS, E and C sublineages) are included except D, for that its last generation within examination scope consist of only 4 cells (samples), which are not enough for the statistics of box plot. Significance level is derived by one-sided Wilcoxon rank-sum test over $n = 128, 16, 8$ and 8 independent cells in AB, MS, E and C (sub)lineages respectively, with $p = 1.4721 \times 10^{-3}$ (MS), 1.2248×10^{-6} (E) and 1.2151×10^{-8} (C), as indicated. The data range between the lower and upper quartiles is illustrated using blue and red boxes for AB and P1 lineages respectively, along with a black line inside indicating the data median. The lower and upper inner fences defined by *Lower Quartile - 1.5 × Interquartile Range* and *Upper Quartile + 1.5 × Interquartile Range* are represented by two black lines extending from the box. The grey points denote the mild outliers smaller than *Lower Quartile - 1.5 × Interquartile Range* or larger than *Upper Quartile + 1.5 × Interquartile Range*. Source data are provided as a Source Data file.

The corresponding sentence is changed into “*As with the division timing, the progeny of the P1 cell are less regularly shaped than those of the AB cell until roughly the 200-cell stage (Figures 4b, Supplementary Figure 8). Regarding the irregularity scores averaged over the cells’ lifespan, the top 10% cells with the most irregular shapes consist of 11 AB progeny and 22 P1 progeny, though the total number of AB progeny is about twice of that of P1 (Supplementary Data 1).*”

Comment 8:

19:25 - since “lamellipodia”, “protrusion” and “filopodia” have distinct meanings, I would say these bumps are “consistent with lamellipodia, protrusions or filopodia”

Response:

The corresponding paragraph is changed into “*These humps, consistent with morphological features of cellular membranes including lamellipodia, protrusion and filopodia, may play a pivotal role in ABpl’s active migration toward the anteroventral direction⁴⁵.*”

Comment 9:

Page 23 - I would also emphasize that since the reason for the decrease in cell segmentation accuracy at later stages is the small size of cells and membrane features relative to the imaging resolution, improved image resolution and quality could allow better performance of CShaper at these stages.

Response:

We now refer the influence of image quality on CShaper's performance - *“Furthermore, improved image quality or novel segmentation algorithm may allow a better performance especially at later stages when cellular size substantially decreases and cells become more crowded.”*